# Learning Tree Interpretation from Object Representation for Deep Reinforcement Learning

**Guiliang Liu**[1,3], **Xiangyu Sun**[2], **Oliver Schulte**[2], **Pascal Poupart**[1,3]
[1]Cheriton School of Computer Science, University of Waterloo
[2]School of Computing Science, Simon Fraser University
[3]Vector Institute
g233liu@uwaterloo.ca, xiangyu_sun@sfu.ca
oschulte@cs.sfu.ca, ppoupart@uwaterloo.ca.

## Abstract

Interpreting Deep Reinforcement Learning (DRL) models is important to enhance trust and comply with transparency regulations. Existing methods typically explain a DRL model by visualizing the importance of low-level *input features* with super-pixels, attentions, or saliency maps. Our approach provides an interpretation based on high-level *latent object features* derived from a disentangled representation. We propose a Represent And Mimic (RAMi) framework for training 1) an *identifiable* latent representation to capture the *independent* factors of variation for the objects and 2) a mimic tree that extracts the causal impact of the latent features on DRL action values. To jointly optimize both the fidelity and the simplicity of a mimic tree, we derive a novel Minimum Description Length (MDL) objective based on the Information Bottleneck (IB) principle. Based on this objective, we describe a Monte Carlo Regression Tree Search (MCRTS) algorithm that explores different splits to find the IB-optimal mimic tree. Experiments show that our mimic tree achieves strong approximation performance with significantly fewer nodes than baseline models. We demonstrate the interpretability of our mimic tree by showing latent traversals, decision rules, causal impacts, and human evaluation results.

## 1 Introduction

Deep neural networks have enabled Reinforcement Learning (RL) agents to extract relevant features from image observations and achieve human-level control [1, 2, 3] by modeling action-value functions. Despite their promising performance, the learned knowledge remains implicit in these black-box neural structures, which hinders understanding the importance of input features and how they influence decisions. Most previous DRL interpretations aimed to visualize attention masks [4, 5] or saliency maps [6, 7] for input states. These interpretations are commonly based on entangled raw features in high-dimensional input space, and some recent studies [8, 9, 10] showed that the generated attention masks are inconsistent for local samples in that very different attention distributions can yield equivalent predictions. Moreover, these point-wise importance maps do not identify the underlying causal relationships between target variables and complex inputs.

An alternative approach for generating post-hoc global explanations is mimic learning [11], which allows distilling the knowledge from an opaque DRL model to a transparent tree model. Some recent works [12, 13] built linear model trees to imitate the DRL action-value functions. These trees are learned for the high-dimensional input space; therefore they require numerous splits for building a tree with promising regression performance. The large tree size increases the complexity of their interpretations and makes it difficult to generate human-understandable decision rules. Previous models [14, 15] constrained the tree complexity by empirically setting some hyper-parameters (e.g,

maximum depth), which, however, limits the mimic performance. It is unclear how to jointly optimize the simplicity and the fidelity of tree models to find a "sweet-spot".

To find an attractive trade-off between simplicity and fidelity, we develop a Represent And Mimic (RAMi) framework based on the Information Bottleneck (IB) principle [16]. The goal of IB is learning a bottleneck representation to compress input signals while maximizing the information about a target variable, which defines *an information-theoretic objective* that views mimic learning as a data compression process. Our novel IB objective facilitates learning a *mimic tree* as the bottleneck to approximate the action advantages from the DRL model with a minimum number of splits. To learn IB-optimal mimic trees for DRL interpretation, RAMi integrates two components.

1) An Identifiable Multi-Object Network (IMONet) that detects objects from high-dimensional input space and learns an object representation to embed the objects with a limited number of latent variables. This representation is a) well-disentangled, meaning that each variable independently captures one factor of variation in a detected object, b) identifiable, allowing the mimic tree to uniquely extract the underlying causal relation from DRL models, and c) interpretable, in order to generate understandable DRL explanations. The interpretability is demonstrated with illustrative examples of latent traversals and human evaluation.

2) A Monte Carlo Regression Tree Search (MCRTS) algorithm that learns mimic trees based on the latent features from the object representation. MCRTS incorporates the IB principle into its search heuristic by deriving a Minimum Description Length (MDL) objective. Unlike previous tree learners that deterministically select a split by evaluating only its local influence, MCRTS looks ahead to global mimic performance and generates a compact distribution of candidate trees with explorations.

Our empirical evaluation shows that IMONet+MCRTS achieves a promising mimic performance with significantly fewer splits than other baseline models and an important reduction in the size of tree interpretation. To demonstrate how the learned mimic tree makes its target action-advantage function interpretable we show causal relations and results of counterfactual interventions can be extracted from it. We validate the tree interpretations by 1) comparison with previous interpretable RL models, and 2) a survey that collects human evaluations.

**Contributions.** 1) We propose a RAMi framework that enables representing the object information with our novel IMONet and searching for the optimal mimic tree with the MCRTS algorithm. 2) We derive an information-theoretic IB-MDL objective that incorporates both the fidelity and simplicity for mimic tree learning. 3) We introduce our method of leveraging the mimic tree to compute feature importance and extract causal relations from a DRL model.

## 2    Represent and Mimic Framework

We propose a Represent And Mimic (RAMi) framework based on the Information Bottleneck (IB) principle. RAMi separately interprets the feature extraction and the decision making process of a DRL model with 1) an interpretable latent representation and 2) a transparent mimic tree.

### 2.1    Mimic Learning for DRL

RAMi applies mimic learning [11] to learn post-hoc interpretations for a DRL model by transferring its knowledge to a mimic tree $\phi$. To facilitate the knowledge distillation, mimic learners utilize soft-outputs from deep models as targets to supervise the training process of mimic trees. DRL models typically compute value functions ($V(s)$ or $Q(s, a)$) to estimate the expected cumulative rewards at state $s$. These value functions explicitly influence the RL decisions by controlling the policy gradient: $Q(s, a)\nabla\pi(a|s)$ [17] or determining the action: $\hat{a} = \arg\max_a Q(s, a)$ [18]. Compared to the policy $\pi(a|s)$, value functions evaluate candidate actions or states more directly. Based on the value functions, one can compute an advantage value for each action by subtracting a baseline: $y = Q(s, a) - V(s)$[1], which creates unbiased action-specific evaluations with less variance. Our mimic learner utilizes action advantages as the mimic targets, allowing us to understand when an action can outperform others by a certain advantage.

---

[1]Action advantages can be defined as $V(s') + r - V(s)$ if Q function is unknown (e.g., in A3C).

## 2.2 IB Method for Representing and Mimicking

We derive an objective for our RAMi framework based on the Information Bottleneck (IB) principle [16]. An ideal mimic tree achieves a promising approximation performance (fidelity) with a minimum number of splits (simplicity). The IB objective naturally integrates both goals, by encouraging a bottleneck representation to compress the input signals $X$ and preserve as much of the relevant information about mimic targets $Y$ (action advantages) as possible. The IB objective is:

$$\max_{\boldsymbol{\omega}} \Big[ I_{\boldsymbol{\omega}}(\Phi, Y) - \lambda I_{\boldsymbol{\omega}}(\Phi, X) \Big] \tag{1}$$

where $I$ denotes the mutual information, and $\lambda$ is a Lagrange multiplier. $-I_{\boldsymbol{\omega}}(\Phi, X)$ controls how much the mimic model compresses the input data, and $I_{\boldsymbol{\omega}}(\Phi, Y)$ measures how well the mimic model preserves information about targets[2]. The IB principle defines what we mean by a good representation, in terms of the fundamental trade-off between having a concise representation and one with good mimic performance [19]. However, in practice, it is difficult to learn a mimic tree from high-dimensional and entangled raw inputs (e.g., images and text embeddings), so we learn a latent object representation from input space and then build a mimic tree upon latent features. Inspired by the deep variational IB [19], The objective for our RAMi framework is given by the following result.

**Theorem 1** *Consider a dataset of size $N$ with input features $\boldsymbol{x}$ and prediction target $y$. Let $\mathcal{L}_q(y_n) = -\log q(y_n|\boldsymbol{\phi})$ denote the description length for encoding the target $y_n$, let $\mathcal{L}_p(\boldsymbol{\phi}) = -\log p_0(\boldsymbol{\phi})$ denote the description length for encoding the mimic model $\boldsymbol{\phi}$. Optimizing the IB objective is equivalent to maximizing its lower bound, which is:*

$$\frac{1}{N} \sum_{n=1}^{N} \Big\{ \mathbb{E}_{q(\boldsymbol{z}|\boldsymbol{x}_n)}[\log p(\boldsymbol{x}_n|\boldsymbol{z})] - \lambda \mathcal{D}_{KL}[q(\boldsymbol{z}|\boldsymbol{x}_n)\|p_0(\boldsymbol{z})] -$$
$$\mathbb{E}_{q(\boldsymbol{z}|\boldsymbol{x}_n)} \Big[ \mathbb{E}_{q(\boldsymbol{\phi}|\boldsymbol{z})} \Big( \mathcal{L}_q(y_n) + \lambda \mathcal{L}_p(\boldsymbol{\phi}) \Big) - \lambda H[q(\boldsymbol{\phi}|\boldsymbol{z})] \Big] \Big\} \tag{2}$$
$$= \textit{ELBo objective} + \textit{IB-MDL objective} + \textit{Entropy Regularizer}$$

The proof can be found in the appendix. This lower bound defines two sub-objectives: 1) the Evidence Lower Bound (ELBo) objective enables learning an object representation for inputs $q(Z|X)$ with our IMONet (Section 3). 2) The IB-MDL objective, for which we design a MCRTS algorithm (Section 4). MCRTS learns a compact distribution over mimic trees $q(\Phi|Z)$ where we select the optimal mimic tree $\boldsymbol{\phi}^*$ to achieve a "sweet-spot" between fidelity and simplicity.

# 3 Learning Object Representation

This section introduces the Identifiable Multi-Object Network (IMONet). The observed inputs from the RL environment are state-action-reward triplets $\boldsymbol{x}_n = [\boldsymbol{s}_n, \boldsymbol{a}_n, r_n]$. Actions and rewards are often well-disentangled in low dimensions, so our IMONet learns a disentangled latent representation for *capturing the object features from states.* We 1) introduce the key properties of IMONet that enable extraction of causal relationships, 2) describe the model structure and 3) discuss the interpretability for the learned latent representation.

## 3.1 Disentangled Representation for Causal Interpretation

The high-dimensional state features from the RL environment are often *entangled* (e.g., highly correlated pixels in an image), which makes causal inference from the raw input space impossible. IMONet utilizes a Variational Auto-Encoder (VAE) framework [20] that converts state features to a disentangled latent representation (i.e., $p(Z) = \prod_{d=1}^{D} p(Z_d)$). After optimizing a generative model for the observed states that utilizes the latent representations, the independent latent variables capture complete state information. The independence implies that there is *no* (unobserved or observed) confounder between two latent variables [21, 22]. This property enables mimic learners to model causal relations between latent variables $Z$ and the target variable $Y$ following a Causal Decision Tree [23]. (Section 5.2 shows examples of causal relations). Since the latent representation encodes complex RL states with few latent variables, the efficiency of causal inference is improved significantly.

---

[2]$X$, $Y$, $Z$, and $\Phi$ are random variables; $\boldsymbol{x}$, $y$, $\boldsymbol{z}$, and $\boldsymbol{\phi}$ are instances of random variables, so $p(X)$ represents a distribution while $p(\boldsymbol{x})$ defines a probability. $F_{\boldsymbol{\omega}}(\cdot)$ is the functional parameterized by $\boldsymbol{\omega}$.

## 3.2 Identifiability of Latent Representation

Identifiability is one of the most fundamental properties for uniquely identifying the causal relations between inputs and targets. If a latent representation is unidentifiable, there exists multiple distributions with the same generative performance, and the variables between two distributions can be completely *entangled*, meaning that they develop inconsistent factorizations for the same inputs [24] (see Section E.1 in appendix for examples). With unidentifiable representations, the mimic learner can return many different causal relations between the latent and target variables.

To better understand this problem, we define identifiability in the unsupervised representation learning setting as follows: $p(Z)$ is identifiable up to equivalence $\sim_B$, if $\forall s \in \mathcal{S}$, $p(\boldsymbol{s}) = \int p(\boldsymbol{s}|\boldsymbol{z})p(\boldsymbol{z})\mathrm{d}\boldsymbol{z}$ $= \int p(\boldsymbol{s}|\boldsymbol{z}')p(\boldsymbol{z}')\mathrm{d}\boldsymbol{z} \implies \exists B, c$ s.t. $T(p(Z)) = BT(p(Z')) + \boldsymbol{c}$, where function $T$ computes sufficient statistics of a distribution, $\boldsymbol{c}$ is a vector, and $B$ is an invertible matrix (that represents a bijective mapping). If $B$ is a permutation matrix, we have identfiability up to a permutation, otherwise up to scaling [25]. Previous work [24] proved that without including an inductive bias (e.g., specific model designs or additional observations), it is impossible to learn identifiable representations without supervising signals. To ensure the identifiability of the latent object representation, IMONet 1) utilizes a Multi-Object Network (MONet) [26] structure that is specially designed to capture variations of objects, and 2) employs a conditionally factored prior $p(Z|A, R) = \prod_d p(Z_d|A, R)$ where each variable $Z_d$ has a univariate Gaussian distribution. Following an Identifiable VAE (IVAE) design [25], IMONet learns a conditional approximate posterior $q(Z|S, A, R)$ as the state representation.

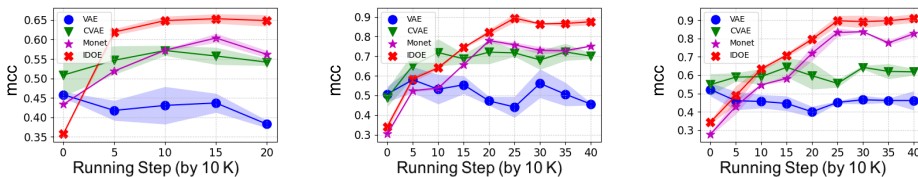

Figure 1: Mean Correlation Coefficients (MCC) for the examined variational encoders (including VAE, Conditional VAE, MONet and IMONet) trained in the Flappy Bird (left), Space Invaders (middle), and Assault (right) environments. We report the mean±variance MCC scores computed from three independent runs.

**Evaluation.** We evaluate the identifiability of IMONet by comparing its Mean Correlation Coefficient (MCC) with other variational encoders. MCC measures whether it is possible to identify latent variables from one model with variables from another model (up to point-wise transformations). To compute MCC, we 1) independently train two encoders based on the same structure, 2) pair latent variables from the encoders, by solving a linear sum assignment problem for the best overall correlation, 3) average the correlation coefficients between paired latent variables at different training steps. The results (Figure 1) demonstrate that conditioning variables and an object network can consistently improve the identifiability of latent representations, whereas VAEs show no evidence of identifiability. This finding is consistent with [24, 25].

## 3.3 Model Implementation

IMONet decomposes a state $\boldsymbol{s}$ into object-level components with spatial masks $(m_1, ..., m_K)$ from a multi-layer convolutional attention network ($m_k$ indicates whether the $k^{th}$ object is present in each pixel, see examples in Figure 2). Conditioning on $m_k$, $r$, and $\boldsymbol{a}$, IMONet independently encodes each component with an encoder-decoder Conditional VAE (CVAE). To train IMONet, we decompose the ELBo from our IB objective (Theorem 1) into an object-oriented variational lower bound:

$$\log \sum_{k=1}^{K} m_k p_d(\boldsymbol{s}|\boldsymbol{z}_k) - \beta \mathcal{D}_{KL}\Big( \prod_{k=1}^{K} q_{enc}(\boldsymbol{z}_k|\boldsymbol{s}, \boldsymbol{a}, r, m_k) \| p(\boldsymbol{z}_k|\boldsymbol{a}, r) \Big) - \sum_{k=1}^{K} \lambda \mathcal{D}_{KL}\Big( q(m_k|\boldsymbol{s}, \boldsymbol{a}, r) \| p_d(m_k|\boldsymbol{z}_k) \Big) \tag{3}$$

where 1) the *first* term is a decoder log-likelihood that defines the reconstruction performance for a mixtures of decoder outputs. The state decoder $p_d(\boldsymbol{s}|\boldsymbol{z}_k)$ is implemented by deconvolution layers. 2) The *second* term is the KL-divergence (KLD) between the conditional approximate posteriors for

each component and a conditional prior. The approximate posterior is implemented by:

$$q_{enc}(\boldsymbol{z}|\boldsymbol{s}, \boldsymbol{a}, r, m_k) := \mathcal{N}(\boldsymbol{\mu}_q, diag(\boldsymbol{\sigma}_q^2)) \text{ s.t. } [\boldsymbol{\mu}_p, \boldsymbol{\sigma}_q^2] = \psi^q[\psi^{conv}(\boldsymbol{s}, m_k), \psi^{mlp}(\boldsymbol{a}, r)] \quad (4)$$

$\psi^{mlp}$ and $\psi^q$ are multi-layer perceptrons while $\psi^{conv}$ denotes a convolutional network. The prior $p(\boldsymbol{z}|\boldsymbol{a}, r)$ is implemented by another Gaussian function without $\psi^{conv}$. 3) The *last* term defines the KLD between the mask generated by the attention network and the mask reconstructed by the CVAE. Our attention network $q(m_k|\boldsymbol{s}, \boldsymbol{a}, r)$ is implemented by U-Net [27], which is implemented a layer-by-layer convolutional neural network with instance normalization, followed by a sigmoid function that transfers logistics to attentions. The mask decoder $p_d(m_k|\boldsymbol{z}_k)$ applies the same structure as our state decoder.

Learning an object representation for the RL states is applicable in practice since 1) a RL state can typically be represented by a limited number of objects, and 2) the agent can interact with the RL environment to generate sufficient data for learning such a representation. Object-Oriented RL [28, 29, 30] has demonstrated that an agent can master RL games by modeling the object dynamics, so we utilize features from the object representation as the inputs to mimic learners: we sample latent features $\boldsymbol{z}_k$ from each component posterior $q_{enc}(Z_k|S = \boldsymbol{s}, A = \boldsymbol{a}, R = r, M = m_k)$, concatenate them with the conditioning values, and feed the entire vector $[\boldsymbol{z}_{1,...,K}, \boldsymbol{a}, r]$ to MCRTS .

### 3.4  Interpreting Latent Variables

IMONet learns a symbolic abstraction of state space by representing object variations (e.g., shapes and locations) with latent variables. A latent variable $Z_{k,d}$ (the $d^{th}$ variable for the $k^{th}$ object) captures an independent factor of object variations. A common approach to reveal the information captured by $Z_{k,d}$ is latent traversing [26, 31, 32]: we randomly choose images and inspect how the reconstruction components change as $Z_{k,d}$ is traversed from (empirical) minimum to maximum values. Figure 2 illustrates the objects captured by attention masks from IMONet and shows two examples of latent traversals in a flappy bird environment. We include a human evaluation to demonstrate the interpretability of latent variables (Section 5.2). The appendix includes more examples (Section E.3 ).

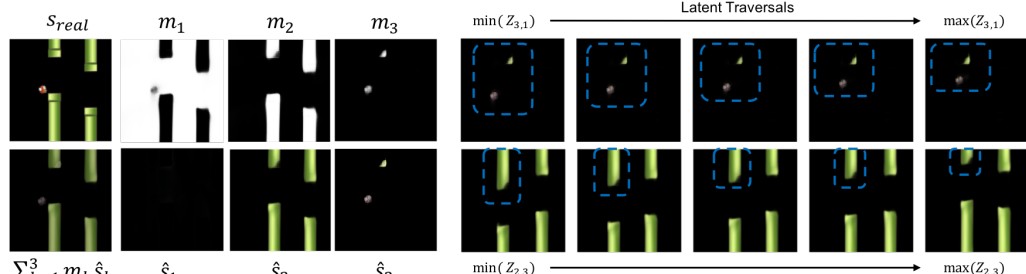

Figure 2: Visualizations for IMONet outputs. IMONet decomposes a state $\boldsymbol{s}_{real}$ into three objects with masks $m_1$ (for the background), $m_2$ (for pillars) and $m_3$ (for the bird), where *white/dark* colors mark *captured/uncaptured* regions. The generations from the decoder are $\hat{\boldsymbol{s}}_1$, $\hat{\boldsymbol{s}}_2$, and $\hat{\boldsymbol{s}}_3$. We show two latent traversals [33] for interpreting $Z_{3,1}$ (the $1^{st}$ variable for the $3^{rd}$ object) and $Z_{2,3}$ (the $3^{rd}$ variable for the $2^{nd}$ object). $Z_{3,1}$ captures the vertical distance between the bird and a pillar, and $Z_{2,3}$ captures the length of the left pillar (highlighted by blue frames).

## 4  Learning Mimic Tree Interpretations

This section introduces the approach to infer a mimic tree with the IB-MDL objective (in Theorem 1) and a Monte Carlo Regression Tree Search (MCRTS) algorithm to find the optimal tree.

### 4.1  Inferring Mimic Trees with IB-MDL

We define $\mathcal{L}_p(\phi)$ and $\mathcal{L}_q(y_n)$ in the IB-MDL objective for mimic tree inference. The goal of MDL is to find regularity and simplicity in the data by viewing learning as a compression problem. There are two agents with complete copies of the latent features $\boldsymbol{z}_n$, but only the sender agent knows the target labels $y_n$. The sender agents transmits to the receiver agent a description of the missing labels using as few bits as possible; we use a mimic tree to encode the information. The optimal mimic tree is

defined to be the one that enables sending the minimum description length of (a) *encoding the tree structure* $\mathcal{L}_p(\phi)$, and (b) *encoding the exceptions* $\mathcal{L}_q(y_n)$ at each leaf node.

**(a)** *Encoding Tree Structure:* We convert the binary tree structure to a string that records the splits ($f$) and leaf predictions ($\hat{y}$) (e.g., $1, f_0, 1, f_1, 0, \hat{y}_0, 0, \hat{y}_1, 0, \hat{y}_2$ where we mark splits and leaves with 1 and 0) with depth-first search [34]. The next proposition gives the encoding cost $\mathcal{L}_p(\phi)$.

**Proposition 1** *Given a regression tree with $L$ splits, the total cost (in bits) of describing the tree structure with the string encoding method is:*

$$\mathcal{L}_p(\phi) = \log \frac{(2L-1)^2}{L^{\frac{3}{2}}(L-1)^{\frac{1}{2}}} + (2L-1)H(\frac{L}{2L-1}) + O(L^{-1}) \tag{5}$$

**(b)** *Encoding the Exceptions:* Traditional MDL tree algorithms [35] are proposed for classification trees to handle discrete labels. Their exception-encoding methods are infeasible with continuous predictions, so we utilize an alternative approach that models the distribution of action advantages at a leaf node $i$ with a Gaussian distribution $\mathcal{N}(\hat{y}_i^{tree}(\phi), \hat{\sigma}_i^2(\phi))$. Given a mimic tree $\phi$, $\hat{y}_i^{tree}$ and $\hat{\sigma}_i^2$ model the prediction and the scale of exception at the leaf $i$ respectively. Inspired by the variance reduction objective in classic regression trees, we utilize the log-variance to measure the cost of encoding exceptions, so $\mathbb{E}_{p(y_n)}[\mathcal{L}_q(y_n)] := \log\left[\sum_{i=0}^{L+1}\frac{N_i^{leaf}}{N}\hat{\sigma}_i^2\right]$ where $N$ and $N_i^{leaf}$ denote the total data size and the number of data points on the $i^{th}$ leaf node respectively.

### 4.2 MCRTS Implementation

MCRTS (Figure 3) takes latent features $\boldsymbol{z}_n \in \mathbb{R}^{K \times D}$ (K and D denote the number of objects and latent size respectively) from our IMONet and the conditioning values $\boldsymbol{a}_n, r_n$ as inputs. It explores different splits by maintaining multiple candidate mimic trees that are trained to approximate action advantages $Y$. To initiate a tree search, MCRTS stores all the instances (data points) in a cell $cell(\langle \boldsymbol{z}_1, \boldsymbol{a}_1, r_1; y_1\rangle, \langle \boldsymbol{z}_2, \boldsymbol{a}_2, r_2; y_2\rangle, \langle \boldsymbol{z}_3, \boldsymbol{a}_3, r_3; y_3\rangle, \dots)$ at the root node. An edge in the *MCRTS search tree* represents a split $f$ in the *mimic tree*. This binary split passes the instances from a parent cell to two partition cells

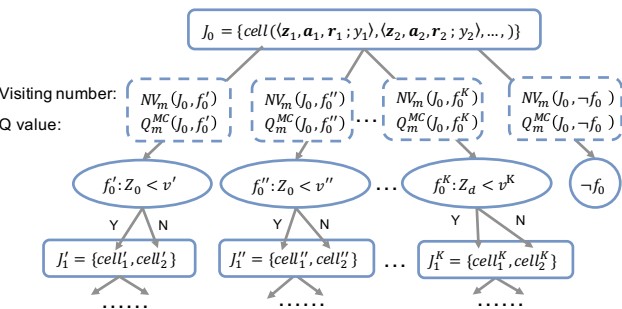

Figure 3: MCRTS structure, where a tree node records partition cells and an edge stores a number of visits $NV$ and an estimate $Q(J, f)$. $f_l^k : Z_d < v$ denotes the $k^{th}$ split at layer $l$, which checks whether $Z_d$ is smaller than a value $v$. The layer number $l$ equals the number of splits in a path. $\neg f_l$ denotes splitting ends.

in child nodes. Each MCRTS edge records a number of visits $NV(J, f)$ and an estimate $Q^{MC}(J, f)$ of the expected splitting influence. Each node $J$ contains a series of partition cells constructed by splits from the root node: $J := \{cell_1(\langle \boldsymbol{z}_1, \boldsymbol{a}_1, r_1; y_1\rangle, \langle \boldsymbol{z}_3, \boldsymbol{a}_3, r_3; y_3\rangle, \dots), cell_2(\langle \boldsymbol{z}_2, \boldsymbol{a}_2, r_2; y_2\rangle, \langle \boldsymbol{z}_4, \boldsymbol{a}_4, r_4; y_4\rangle, \dots), cell_3(\dots)\}$. MCRTS learns a compact distribution of mimic trees $p(\Phi|Z)$. A *mimic tree $\phi \in \Phi$ can be extracted from MCRTS by following a path from the root to a leaf node $J_l$, so $\phi := \{f_0 \dots f_{l-1}, J_l\}$. A partition $cell_i$ in $J_l$ defines a leaf node in the mimic tree $\phi$, and the leaf prediction is $\hat{y}_i^{tree} = \sum_{y \in cell_i} y/N_i^{leaf}$.* Appendix (E.1) provides a *mimic tree extraction* example.

**Searching:** MCRTS implements a tree search by running $M$ *plays* from a starting node $J_s$ (which is initialized to be the root node and updated after each *move*) with root parallelization [36]. At the $m^{th}$ play, MCRTS implements four phases including:

1) *Selection:* Traverse the tree from $J_s$ to a leaf node by selecting the split $f_{l,m}$ at each layer to maximize the Upper Confidence Bound (UCB) [37]:

$$f_{l,m} = \arg\max_f \left[ Q_{m-1}^{MC}(J_l, f) + c_{puct}\frac{\sqrt{\log(m-1)}}{NV_{m-1}(J_l, f)+1} \right] \tag{6}$$

where $c_{puct}$ controls the scale of exploration. Similar to the entropy regularizer (Theorem 1), a large $c_{puct}$ prevents over-frequent visits to a node. *2) Evaluation:* Evaluate the selected leaf $J_{leaf}$ with reward: $r^{MC}(J_{leaf}) = -\mathcal{L}_q(y_n) - \lambda\mathcal{L}_p(\phi)$. *3) Expansion:* Expand the leaf node

Table 1: Mimic performance. We use the Variance Reduction (VR) and Variance Reduction Per Leaf (VR-PL) metrics since they are common regression objectives and optimized by the IB-MDL objective. Results for other regression metrics (Root Mean Square Error and Mean Absolute Error) and the corresponding variances are recorded in Table B.2 in appendix. $w_+$ indicates that each leaf node has an extra linear regression model. We omit the results for the *raw-input-based MCRTS* since it is computational intractable.

| | Flappy Bird | | | Space Invaders | | | Assault | | |
|---|---|---|---|---|---|---|---|---|---|
| Method | VR | VR-PL | Leaf | VR | VR-PL | Leaf | VR | VR-PL | Leaf |
| Cart | 8.51E-2 | 8.43E-5 | 1007 | 4.96E-2 | 7.02E-5 | 705 | 4.79E-2 | 7.46E-5 | 642 |
| VIPER | 8.57E-2 | 1.88E-4 | 453 | 4.63E-2 | 8.80E-5 | 525 | 5.28E-2 | 8.09E-5 | 653 |
| M5-RT | **9.59E-2** | 8.37E-5 | 1144 | 4.54E-2 | 2.92E-5 | 1558 | 4.37E-2 | 2.73E-5 | 1605 |
| M5-MT | 9.56E-2 | 1.55E-4 | $612^{w+}$ | 1.60E-2 | 1.23E-5 | $1303^{w+}$ | 3.42E-2 | 2.54E-5 | $1351^{w+}$ |
| GM-LMT | 8.99E-2 | 2.99E-4 | $303^{w+}$ | 2.07E-2 | 8.32E-5 | $249^{w+}$ | 5.55E-2 | 1.83E-4 | $307^{w+}$ |
| VR-LMT | 8.46E-2 | 5.36E-4 | $157^{w+}$ | 2.65E-2 | 1.61E-4 | $166^{w+}$ | 5.80E-2 | 1.98E-4 | $291^{w+}$ |
| VAE+CART | 7.25E-2 | 3.44E-4 | 212 | 3.99E-2 | 7.86E-5 | 507 | 5.15E-2 | 1.16E-4 | 448 |
| VAE+VIPER | 7.63E-2 | 5.32E-4 | 143 | 4.12E-2 | 9.89E-5 | 417 | 4.57E-2 | 1.29E-4 | 356 |
| VAE+GM-LMT | 6.35E-2 | 3.51E-4 | $180^{w+}$ | 3.39E-2 | 2.75E-4 | $123^{w+}$ | 4.20E-2 | 1.44E-5 | $293^{w+}$ |
| VAE+VR-LMT | 7.95E-2 | 5.12E-4 | $154^{w+}$ | 3.52E-2 | 2.08E-4 | $171^{w+}$ | 5.10E-2 | 1.99E-4 | $258^{w+}$ |
| VAE+MCRTS | 7.83E-2 | 1.27E-3 | 61 | 4.82E-2 | 5.66E-4 | 85 | 6.58E-2 | 7.75E-4 | 85 |
| IMONet+CART | 8.23E-2 | 4.02E-4 | 204 | 5.21E-2 | 1.38E-4 | 375 | 5.67E-2 | 1.81E-4 | 315 |
| IMONet+VIPER | 8.50E-2 | 4.48E-4 | 191 | 5.26E-2 | 1.69E-4 | 313 | 6.05E-2 | 1.90E-4 | 319 |
| IMONet+GM-LMT | 7.87E-2 | 3.74E-4 | $212^{w+}$ | 4.79E-2 | 3.23E-4 | $149^{w+}$ | 5.45E-2 | 2.15E-4 | $256^{w+}$ |
| IMONet+VR-LMT | 8.21E-2 | 7.16E-4 | $115^{w+}$ | 4.54E-2 | 3.79E-4 | $120^{w+}$ | 6.03E-2 | 2.27E-4 | $268^{w+}$ |
| IMONet+MCRTS | 8.53E-2 | **1.37E-3** | **62** | **5.37E-2** | **7.08E-4** | **76** | **7.53E-2** | **9.07E-4** | **83** |

with $G$ (the maximum exploration width) child nodes. *4) Back Up:* Update the action-values: $Q_m^{MC} = (Q_{m-1}^{MC} NV_{m-1} + r^{MC})/(NV_{m-1} + 1)$ and increment the visit count: $NV_m = NV_{m-1} + 1$ on all the traversed edges.

**Move:** After $M$ plays, we *move* $J_s$ to a child node by selecting the split $\tilde{f}_s = \arg\max_{f_k \in F_{1\ldots g(m)}} NV_m(J_s, f_k)$ and setting $J_s$ to a child node $\tilde{J}_{s+1}$ (connected by the edge $\tilde{f}_s$). The next play will start from the new starting node $\tilde{J}_{s+1}$. It allows MCRTS to prune the sub-optimal nodes [38].

## 5 Empirical Evaluation

We evaluate the mimic performance and demonstrate the interpretability of the mimic tree.

**Environment and Running Settings:** We study the Flappy Bird, Space Invaders, and Assaults environments. Flappy Bird is a procedural game, where the game states are randomly generated at each episode. Space Invaders and Assaults are commonly studied Atari games from the Gym toolkit [39]. (Check *data generation details and model hyper-parameters* in Appendix).

**Baseline Models:** We compare previous tree-based mimic learners. The first baseline model is a Classification and Regression Tree (**CART** ) [40]. We include **VIPER** [41] as our second baseline by replacing its policy (decision) tree with a regression tree that imitates action values with a Q-dagger algorithm. The third baseline is the M5 [42] regression tree training algorithm. M5 constructs a piecewise constant tree by pruning the sub-optimal nodes. We include M5 with the Regression-Tree option (**M5-RT**) and the Model-Tree option (**M5-MT**). M5-MT builds a linear regression model for the instances at each leaf node while M5-RT maintains only a constant value. The last baseline is a Linear Model Tree (LMT) [12] for interpreting action values. A recent work [13] explored different heuristics for selecting splitting features, including Variance Reduction (**VR-LMT**) and Gaussian Mixture clustering (**GM-LMT**). The aforementioned models are directly learnt from raw input space (states and actions) [14]. The aforementioned models were previously learnt directly from raw input space (states and actions). In this study of latent representation, we evaluate their regression performance based on the latent features from both VAE and IMONet (Check appendix for the implementation details of baseline models).

### 5.1 Fidelity versus Simplicity

This experiment evaluates mimic trees by their 1) fidelity: how well a mimic model approximates the DRL model [12] and 2) simplicity: the size of these nonparametric trees. We divide the dataset (50k) into training (80%), validation (10%), and testing (10%) sets and generate 5 independent runs. Fidelity and simplicity are evaluated by the mimic performance and the number of leaves respectively.

Table 1 shows the regression performance. MCRTS+IMONet achieves a promising regression performance with significantly fewer leaves than other baseline models. This is because 1) IMONet learns a concise object representation from RL states and this representation outperforms that modelled by an unidentifiable VAE. 2) MCRTS selects splits to maximize the regression performance at a global level, yielding a mimic tree with better fidelity. Apart from these obvious advantages, we find that some latent features may be indistinguishable for predicting action values, and this explains why some trees have better regression performance by building a large tree from raw inputs, while the large tree size compromises their interpretability.

**Leaf-by-Leaf Regression Performance:** To study the regression efficiency, we evaluate the per-leaf regression performance of examined models based on the latent features from IMONet. Figure 4 illustrates the performance of examined mimic trees when we constrain their number of leaves. MCRTS achieves a leading regression performance, as it selects a split by looking ahead to the future cumulative rewards instead of its local influence. Another key observation is that a well-explored split from MCRTS can beat a greedy split with extra linear regressors at leaf nodes (e.g., LMTs), allowing MCRTS to substantially improve the learning efficiency.

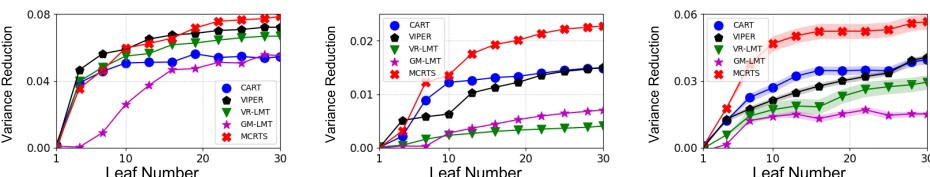

Figure 4: Leaf-by-leaf tree regression results based on latent features from IMONet in the Flappy Bird (left), Space Invaders (middle), and Assault (right) environments.

## 5.2 Illustrative Examples of Interpretable Mimic Trees

We demonstrate the interpretability of our mimic tree by 1) illustrating the extracted rules and causal relations, 2) comparing with previous RL interpretations and 3) conducting human evaluations. This section uses the Flappy Bird environment as an example, where a DRL agent earns rewards by controlling a bird to pass pillars.

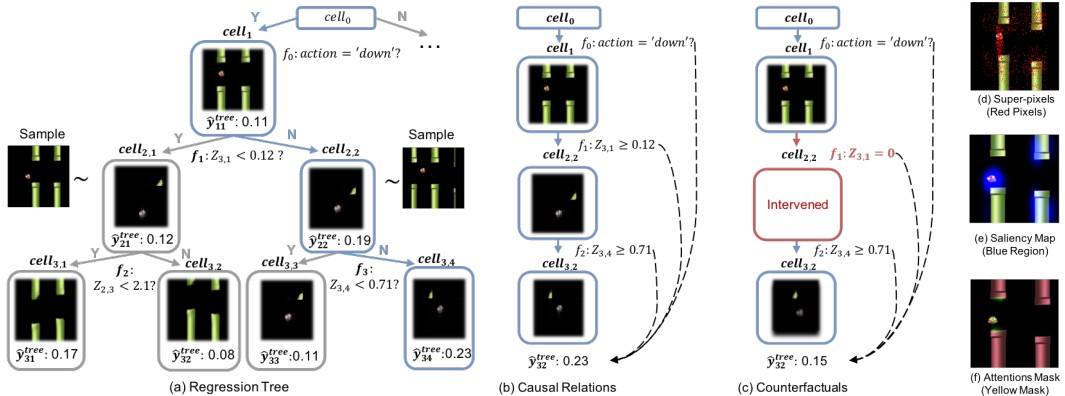

Figure 5: Mimic tree. $f_l : Z_{k,d} < v$ indicates the $l^{th}$ tree split is at $Z_{k,d}$ ($d^{th}$ variable for the $k^{th}$ object) with a splitting value $v$. In the decision rules (b&c), *solid / dash* lines indicate *tree path / causal relations*.

**Causal Explanations:** Our mimic tree $\phi$ captures the influence of latent features on action advantages $y$. Figure 5 (a) illustrates the top three layers of the mimic tree. We can interpret the influence of a split by visualizing the expected latent values after this split. For example, after implementing the split $f_1$, we obtain two child nodes ($cell_{2,1}$ and $cell_{2,2}$). We compute the expected latent values for the instances on these nodes ($\mathbb{E}_{n \in cell}(\boldsymbol{Z}_n)$) and visualize the $3^{rd}$ object with the decoder (since $Z_{3,1}$ captures the variation of the $3^{rd}$ object). These visualizations capture the difference of object information after splitting. They show that the instances on $cell_{2,2}$ have shorter (vertical) distance

between the bird and the upper pillar, and thus their advantages for the action 'down' are larger. We verify this observation by sampling an instance from both child nodes respectively. This observation is consistent with game rules since performing the action 'down' when the bird approaches the upper pillar can prevent potential crashes and thus has a larger advantage.

**(a)** *Causal Relation Extraction.* Since there is no (observed/unobserved) confounder between latent variables (section 3.1), our mimic tree can capture causal relations between splitting features and predictions. Similar to Causal Decision Trees (CDTs) [23], a tree split $f$ in our mimic tree represents a *context-specific causal relationship* between the splitting condition $f$ and the action advantage $\hat{y}_i^{tree}$. The context is a series of value assignments of the latent features along the path from the root to the parent node of $f$. For example, the decision rule in Figure 5(b) states that having $Z_{3,4} \geq 0.71$ is casually related to an action advantage of 0.23 when the action is 'down' and $Z_{3,1} \geq 0.12$ (context).

**(b)** *Counterfactual Analysis.* Leveraging the generation ability of the IMONet decoder, our mimic tree enables counterfactual analysis to answer "what if" questions: we show the influence on tree prediction by intervening to set the value of $Z_{k,d}$ equal to some particular value, For example, in Figure 5(c), we compute $F_{tree}[f_0, f_1, \ldots, f_L, X', Y'_{\theta} | \text{do}(Z_{3,1} = 0)]$. To achieve this, 1) we set $Z_{3,1}$ to 0 for all input latent vectors and get $z'$. 2) The IMONet decoder computes $p_d(x'|z')$ and the DRL model predicts $Y'_{\theta}$ with the generated observations $X'$. 3) The mimic tree follows previous splits to construct tree predictions (the splits on the intervened variables are removed). The difference of tree prediction before and after the intervention (0.23 versus 0.11) reveals the influence of $\text{do}(Z_{3,1} = 0)$.

**Comparison to other interpretations.** We compare other interpretable DRL models based on 1) Super-pixels [12] (Figure 5d) that highlight the pixels used as splitting features in a regression tree, 2) Saliency Map [6] (Figure 5e) which is generated by perturbing the pixels within a region and evaluating their impact on target outputs, and 3) Attention Mask [4] (Figure 5f) (from the convolutional attention network) that represents the importance of pixels on model outputs. These traditional models used the raw images as input during training. The learned interpretations visualize the pixels that are important for decisions. They generally agree that the distance between the bird and the pillar is most influential. This observation is consistent with rules extracted by our method (Figure 5a&b) since this distance determines how likely the bird can pass the pillars to win more rewards.

**Human Evaluation.** We evaluate the interpretability of the aforementioned methods with RL practitioners from four universities. We first introduce the saliency map, mimic trees based on super-pixels and latent variables, and then ask the participants to 1) describe the information captured by these methods and 2) rank the corresponding interpretations. To eliminate potential bias, participants are interviewed independently, and they are not told which methods are the baselines or the proposed model (check Section D in appendix for further details). We present a summary of the results by discussing the two aspects of interpretability regarding our method:

**(a)** *Interpretability of Latent Variables.* After observing the latent traversals, all (12/12) participants correctly recognized the physical meaning of the latent variables whereas 58%(7/12) and 33.3%(4/12) participants managed to recognize the features captured by saliency maps and super-pixels due to the inconsistency of local samples and the complexity of tree paths.

**(b)** *Interpretability of mimic trees.* 83% (10/12) participants preferred the mimic tree based on latent features since its tree path captures globally-consistent knowledge and its tree size is smaller than that of the super-pixel tree (62 versus 1007, check Table 1). The rest voted for the saliency map because they preferred straight-forward explanations.

## 5.3 Limitations

**Computational Complexity.** Since MCRTS conducts multiple simulations to explore and generate the global-optimal mimic tree, it generally consumes more computational resources than classic regression. We compare the computational complexity of our method and the baselines in the appendix. Although IMONet compresses high-dimensional inputs into latent vectors and we parallelize the tree search in MCRTS, empirically, it takes MCRTS a longer to determine a split.

**Guarantee for Interpretability.** It is generally hard to theoretically justify or numerically quantify the level of interpretability, so we utilize illustrative examples and human evaluation to empirically

show the interpretability. Although previous works demonstrated interpretability of latent features [26, 31, 32], there is no guarantee that these features always have a clear meaning across all RL games.

## 6    Related Works

We introduce previous works for interpreting DRL models.

**Self-Explainable Approaches:** Some recent works have explored self-explainable DRL models. To derive human-understandable policies, these models modified existing performance-driven DRL models (e.g., DQN) according to some specific designs, such as symbolic planning [43], decision trees [44, 45], Generalized Value Functions (GVFs) [46], Moore Machine Networks (MMNs) [47], transparent decision dynamics [48] or explainable feature extraction [4, 49]. These modifications influence the decision process of DRL agents to enhance interpretability.

**Post-hoc Interpretation:** An alternative approach is to generate posterior explanations for DRL models to achieve post-hoc interpretability. Existing methods commonly apply feature perturbation to generate saliency maps [6, 7] or learn attention masks [5, 50] with the attention neural network. These explanations are driven locally, based on computing a localization map that highlights the important regions for current actions. It is hard for these point-wise interpretations to maintain their consistency with explanations from other samples, which leads to unstable explanations [10]. Mimic learning (i.e., model distillation) [11] enables the generation of globally consistent explanations from a performance-driven DRL model. Previous works [13, 12] utilize mimic regression trees to interpret action values from DRL models, but their training objective did not control the model complexity and thus encouraged oversized mimic trees. Another recent work [51] proposed a tree regularizer that adjusts the loss function of deep models by integrating the regression performance of mimic trees. A recently proposed VIPER model [41] designed a Q-dagger algorithm to derive a tree-based policy function following online imitation learning [52]. Our experiments evaluate an updated VIPER model for estimating action advantages.

## 7    Conclusion and Societal Impacts

This paper introduced a RAMi framework based on the IB principle, following which we 1) introduced an IMONet to learn an object representation for RL states and 2) proposed a MCRTS algorithm to find a mimic tree with promising fidelity and simplicity. Experiments demonstrated the leading mimic performance of our MCRTS+IMONet model. We illustrated our tree interpretations with causal relations and human evaluation. A potential *negative societal impact* of our work is that it encourages the government to endorse laws that regulate the level of transparency in AI systems. This may increase the difficulty of deploying advanced AI systems.

### Acknowledgments and Disclosure of Funding

This project was supported by a Strategic Project Grant from the Natural Sciences and Engineering Research Council of Canada and MITACS Research Training funding. Our computations were facilitated by a GPU donation from NVIDIA. Resources used in preparing this research were provided, in part, by the Province of Ontario, the Government of Canada through CIFAR, and companies sponsoring the Vector Institute.

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
