# Appendix

## A  Experiment Details

### A.1  Environment Details and Game Rules

We exam our model under three environment, including Flappy Bird, Space Invaders and Assault. We select these environments because they are well-studied RL environments with high-dimensional input space. Flappy Bird is procedural game programmed with the pygame package [1], where the states are *generated randomly*. Both the Space Invaders and the Assault are Atari games simulated by OpenAI Gym toolkit [2]. The detailed games rules are:

**Flappy Bird:**   The agent controls a bird to pass the pillars. The pillars are generated randomly by the environment. At each time step, the environment generates a 0.1 reward unless 1) the bird manages to pass a pipe, so a +1 reward will be returned, or 2) the bird hits the pillars or the ground, so a -1 reward will be returned and the game will end.

**Space Invaders:**   The agent controls a spaceship (in green color at the bottom of the screen) to shoot bullets and defeat the aliens (in yellow color at the top). The aliens can move vertically and horizontally or shoot bullets back to the spaceship. To win the game, the spaceship must hide from the aliens' attack. The barrier (red color in the middle) can be destroyed by the bullets. The environment returns a +1 reward when an alien is destroyed.

**Assault:**   The agent must control a spaceship (at the bottom of the screen) to destroy an alien mothership by shooting bullets. The monthership can defend itself by producing small aliens that will gradually move toward the spaceship. The spaceship can lose its life when it is attacked by the aliens. A +1 reward is generated when the agent manages to destroy an alien.

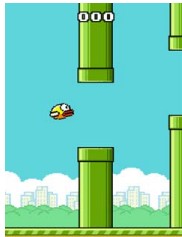 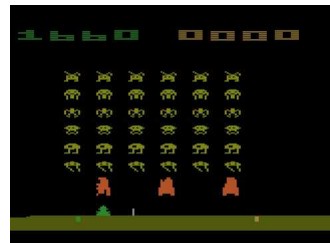 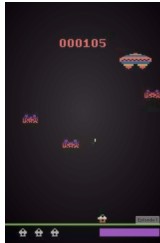

Figure A.1: The DRL environments in our experiment: Flappy Bird (left), Space Invaders (middle), and Assault (right).

35th Conference on Neural Information Processing Systems (NeurIPS 2021).

## A.2    Implementation Details

### A.2.1    DRL Models.

We first train a DRL agent for the above environments. The Flappy Bird agent implements a deep Q-learning model following [3]. The Space Invaders and Assault agents applies the Asynchronous Advantage Actor-Critic (A3C) algorithm  [4]. To ensure the trained models can provide promising action values, we utilize the implementations in a third-party python package named *Tensorpack*. A detailed introduction to the *code and performance* can be found online (URL is temporally omitted due to anonymous regulations).

### A.2.2    Mimic Dataset.

Given the pre-trained DRL model, we collect data by implementing an epsilon-greedy ($\epsilon = 0.01$) policy and store the selected actions and traversed states. We collect a total of 50k data points in the format of ($\langle s_n, a_n, r_n \rangle, y_n$) for both environments. The size of training, validation, and testing set take 80%, 10%, and 10% of total data. Given the large data size, our experiment applies a hold-out validation technique, where we train mimic models with the training set, adjust their hyper-parameters applying the validation set, and test the model performance with the testing set. We implement 5 independent runs and report the mean±variance performance in table C.1.

### A.2.3    Mimic Models.

We provide a detailed introduction to the baseline models:

**Classification And Regression Tree (CART)** is implemented by a third-party python package named Scikit-Learn. Without applying latent features from DE, CART directly selects splitting values from continuous image pixels and utilizes the means of action advantages on leaf nodes as predictions. To prevent over-fitting, we adjust a parameter (set as 10) controlling the minimum number of data points at a leaf node.

**Verifiability via Iterative Policy ExtRaction (VIPER)** [5] is an imitation-based algorithm that learns decision tree policies under the guidance of a DNN policy and its Q-function. To improve the fidelity of the tree policy, VIPER introduces a Q-Dagger algorithm that utilizes cumulative rewards (Q values) from DRL model to resample from the imitation dataset. In this work, to generate fair comparison to other baselines, we replace the policy (decision) tree with an advantage value (regression) tree for selecting actions.

**M5** trees are implemented with a third-party WEKA toolkit [6]. Their implementation follows a previous work [7]. The minimum number of data points (at a leaf node) is set to 10 and 25 for the regression tree (M5-RT) and the model tree (M5-MT). The WEKA toolkit does not allow setting a maximum leaf node, so our leaf-by-leaf results (Figure 5) omit the M5-based methods.

**Linear Model Trees (LMT)** follows the implementation of [8] and applies their source codes (URL is temporally omitted due to anonymous regulations). The minimum number of data points (at a leaf node) is set to 10. In this work, we experiment a total of four heuristics including T-Test (TT), Segment Regression (SR), Gaussian Mixture (GM), and Variance Reduction (VR) heuristics. However, preliminary experiments show that the TT and SR heuristics fail to achieve satisfying performance in both environments, so we omit them in the main paper.

**Monte Carlo Regression Tree Search (MCRTS )** applies a $\lambda$ to control the scale of tree encoding cost $\mathcal{L}_P(\phi)$ in our reward function. We set $\lambda$ to 0.05 for both environments. The play number $N$ is set to 200 and the maximum exploration width $K$ is set to 10. $c_{puct}$ is set to 0.1 and 0.01, 0.01 for the Flappy Bird, Space Invaders and Assault environments. The parameters are determined experimentally.

## A.3    Hyper-parameters

The total number of training data points is 50K ($N = 50,000$). An image observation contains a total of $H = 49,152(128 \times 128 \times 3)$ pixels. For the majority of the RL environments, the number of objects in a state is consistent throughout a game. To determine to object number, we empirically locate the range of K by observing RL states and then determine the smallest K that provides a

satisfying reconstruction performance, since increasing K will expand the dimension of learned latent representation and thus add complexity to the following tree learning. In the experiments, we set the number of objects $K = 3$ and a latent representation contains $D = 16$ dimensions. For implementing the I-MONet, we set the both $\lambda$ and $\beta$ to 0.5 by following [9].

### A.3.1 Computational Complexity

We provide a brief analysis of the computational complexity of examined models.

**CART** applies a greedy search that directly selects splitting features from raw observations. The time complexity of computing a variance is $O(N)$, and thus the time complexity of performing a greedy search is $O(HN^2)$, so constructing a regression tree with $L_{cart}$ splits costs $O(L_{cart}HN^2)$.

**VIPER** iteratively trains multiple trees for action selection and uses the tree with the highest fidelity as the mimic tree to be the DRL interpretation. The training process follows that of CART, so the cost of generating $G$ trees with $L_{viper}$ splits is $O(L_{viper}GHN^2)$.

**M5** algorithm applies a similar tree splitting method. The computational complexity of M5-RT is similar to that of CART. M5-MT adds a linear model at each layer and the time complexity of linear regression with $H$ weights is $O(N^2H + H^3)$, so the time complexity of building M5-MT with $L_{m5}$ splits is $O[L_{m5}(N^2H + H^3) + L_{m5}HN^2]$.

**VR-LMT** applies a sorting variance reduction heuristic for greedy search with complexity $O(N \log N)$, so the total time complexity of VR-LMT with $L_{lmt}$ splits is $O[L_{lmt}(N^2H + H^3) + L_{lmt}HN \log N]$. **GM-LMT** requires building a Gaussian mixture model at each scan which is generally more expensive than computing variance reduction.

**MCRTS** (we visualize the searching phases in Figure A.2) applies latent features from the disentangled representation with $D$ dimensions. At each play, MCRTS traverses to a leaf node by following a UCB heuristic and expands the leaf node with $G$ child nodes, which gives a computational complexity $O(L_{play} + KDN \log N + G)$. To construct a tree with $L_{MCRTS}$ splits, it requires a total cost of $O[L_{MCRTS} M(L_{play} + KDN \log N + G)]$.

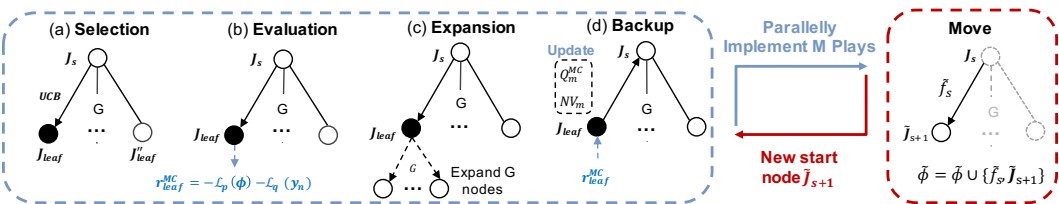

Figure A.2: Monte Carlo Regression Tree Search (MCRTS ) Implementation.

Compared to traditional regression trees, MCRTS applies $M$ Monte Carlo simulations to determine a split, which is more computationally expensive than the greedy method. However, instead of scanning over the image space, MCRTS utilizes latent features whose dimensions are over 1k times smaller than that of raw observations ($KD << H$). To accelerate running, the Monte Carlo simulations are performed simultaneously with root parallelization [10]. In practice, these techniques significantly improve the training efficiency of MCRTS .

**Empirically Comparison:** We do not cover empirically comparisons of the running time for our task since our MCRTS parallelizes the tree search. The running time of MCRTS is correlated to the thread number, which, however, will not influence the speed of traditional regression trees training. It's hard to design an experiment to provide a fair comparison among examined models.

### A.3.2 Latent Traversals

We explain the process of generating latent traversals to interpret the latent variables in the object representation. To generate latent traversals 1) We randomly select 1k images, generate their latent features with IDOE, and compute a latent vector $z_{avg} = \{z_{1,1...D}, z_{2,1...D}, \ldots, z_{K,,1...D}\}$ by averaging all the values of latent variables. The dimension of latent vector is $K \times D$ since the object representation captures the variations of a total of $K$ objects, and it encodes each object with a vector of dimension $D$. 2) To traverse the $d^{th}$ variable for the $k^{th}$ object $Z_{k,d}$, we fix other $D \times K - 1$

values and assign $Z_{k,d}$ from its minimum and maximum values. For each assigned value $z'_{k,d}$, we concatenate it with the rest $K \times D - 1$ latent features and apply the IDOE decoder to generate an image. By observing the variations of generated images, one can recognize the meaning of $Z_{k,d}$.

### A.3.3 Decision Rules Visualization

We visualize the decision rules to interpret DRL models. The tree split is based on latent features. To visualize the influence of a split, we 1) compute the mean latent vectors $\boldsymbol{z}_{avg,lc}$ and $\boldsymbol{z}_{avg,rc}$ at the left and right child nodes respectively, and 2) input $\boldsymbol{z}_{avg,lc}$ and $\boldsymbol{z}_{avg,rc}$ into the decoder to generate images $\boldsymbol{x}_{lc}$ and $\boldsymbol{x}_{rc}$. If the split is on the representation for the $k^{th}$ object (e.g., on $Z_{k,d}$), we condition the image generation process on the mask $m_k$ and visualize only the the $k^{th}$ object. This *interpretation through generation* approach utilizes the rules learned by the decision tree and the generation ability of an IDE decoder, which significantly facilitates understanding the knowledge learned by the mimic tree.

### A.3.4 Experiment Equipment and infrastructures.

Our IDE and other baseline models are trained on a local machine with a TITAN X (Pascal) GPU with 16 GB memory and an Intel Core 7 CPU with 32 GB main memory. MCRTS is implemented on a heterogeneous cluster (named Cedar) belonging to Compute Canada. We apply a computational resource of 64 GB main memory with 16 CPUs at each task.

# B  Theorem Proofs

## B.1  Notations and Definitions

Table B.1 shows a short conclusion of the definitions that are applied in this paper. The following proof will be based on these definitions.

Table B.1: Notations. $q(\cdot)$ denotes the approximate posterior.

| Notation | Explanations |
|---|---|
| $S, A, R$ | State, action, and reward variables in DRL. |
| $Q(\boldsymbol{s}, \boldsymbol{a}), V(\boldsymbol{s})$ | Value functions in DRL. |
| $\Phi$ | Variables for Mimic regression trees |
| $Z$ | Latent variables for a disentangled representation. |
| $X$ | Input variables, $X = (S, A, R)$. |
| $Y$ | Output variable (denotes action advantages). |
| $\boldsymbol{\omega}$ | Parameters of a Mimic Model ($\boldsymbol{\omega} = [\boldsymbol{\omega}_{DE}, \boldsymbol{\omega}_{MC}]$). |
| $p(Y|X; \boldsymbol{\theta})$ | The distribution for a DRL model ($\boldsymbol{\theta}$) outputs. |
| $q(Z|X; \boldsymbol{\omega}_{DE})$ | The disentangled representation modelled by IODE. |
| $p(\Phi|Z; \boldsymbol{\omega}_{MC})$ | The mimic representation modelled by MCRTS . |
| $p_0(\cdot)$ | Priors for Latent distributions. |
| $\lambda$ | A Lagrange multiplier control the scale of compression. |
| $J$ | A MCRTS node. |
| $f$ | A MCRTS spilt. |
| $Q^{MC}$ | Q functions in MCRTS . |
| $NV$ | Visiting number in MCRTS . |
| $c_{puct}$ | The exploration rate in MCRTS . |
| $N$ | The total number of training data points. |
| $M$ | The total number of plays. |
| $K$ | The maximum exploration width. |
| $G$ | The dimension number for observations. |
| $D$ | The dimension number for a disentangled representation. |

## B.2 Proof of Theorem 1

**Proof.** This proof is based on some important intermediate results about the Deep Variational Information Bottleneck [11]. Assuming we have Markov Chain $Y \leftrightarrow X \leftrightarrow \Phi$, we have $p(X, Y, \Phi) = p(Y|X)p(\Phi|X)p(X)$, [11] has proved:

$$I(\Phi; Y) \geq \int p(\boldsymbol{\phi}, y) \log q(y|\boldsymbol{\phi}) \mathrm{d}y \mathrm{d}\boldsymbol{\phi} - H(Y) \tag{1}$$

$$I(\Phi; X) \leq \int p(\boldsymbol{x})p(\boldsymbol{\phi}|\boldsymbol{x})[\log p(\boldsymbol{\phi}|\boldsymbol{x}) - \log p_0(\boldsymbol{\phi})] \mathrm{d}\boldsymbol{x} \mathrm{d}\boldsymbol{\phi} \tag{2}$$

Where $q(y|\boldsymbol{\phi})$ is a variational approximation to $p(y|\boldsymbol{\phi}) = \int \mathrm{p}(y|\boldsymbol{x})p(\boldsymbol{x}|\boldsymbol{z})d\boldsymbol{x}$ and $p_0(\boldsymbol{\phi})$ is a variational approximation to the marginal distribution of $\Phi$, which is $\int p(\boldsymbol{\phi}|\boldsymbol{x})p(\boldsymbol{x})\mathrm{d}\boldsymbol{x}$.

By combining formula (1) and formula (2), we can get

$$I(\Phi; Y) - \lambda I(\Phi; X) \tag{3}$$
$$\geq \int p(\boldsymbol{\phi}, y) \log q(y|\boldsymbol{\phi}) \mathrm{d}y \mathrm{d}\boldsymbol{\phi} - \lambda \Big[ \int p(\boldsymbol{x})p(\boldsymbol{\phi}|\boldsymbol{x})[\log p(\boldsymbol{\phi}|\boldsymbol{x}) - \log p_0(\boldsymbol{\phi})] \mathrm{d}\boldsymbol{x} \mathrm{d}\boldsymbol{\phi} \Big]$$

Note that we omit the entropy of our labels $H(Y)$ because it is independent of our optimization procedure. In practical, we can also approximate $p(\boldsymbol{x}, y) = p(\boldsymbol{x})p(y|\boldsymbol{x})$ using the empirical data distribution, so $p(\boldsymbol{x}, y) = 1/N \sum_{n=1}^{N} \delta_{y_n}(y)\delta_{\boldsymbol{\phi}_n}(\boldsymbol{\phi})$. Based on this empirical approximation, [11] proved the lower bound of IB objective is (or $I(\Phi; Y) - \lambda I(\Phi; X) \geq$):

$$\frac{1}{N} \sum_{n=1}^{N} \int \Big[ p(\boldsymbol{\phi}|\boldsymbol{x}_n) \log q(y_n|\boldsymbol{\phi}) - \lambda \mathcal{D}_{KL}[q(\boldsymbol{\phi}|\boldsymbol{x}_n)||p_0(\boldsymbol{\phi})] \Big] \mathrm{d}\boldsymbol{\phi} \tag{4}$$

However, in practice, the dimension of the input X could be huge and X may contains highly entangle features (e.g., pixels): $p(X) = \prod_{g=1}^{G} p(X_g|X_1, \dots, X_{g-1}, X_{g+1}, \dots, X_G)$. It is difficult to learn interpretable mimic trees from raw input. To enhance interpretability, We first transfer $\boldsymbol{x}_n$ into disentangled latent features $\boldsymbol{z}_n$ by applying our IDE, so $q(\boldsymbol{\phi}|\boldsymbol{x}_n) = \int q(\boldsymbol{\phi}, \boldsymbol{z}|\boldsymbol{x}_n)\mathrm{d}\boldsymbol{z} = \int q(\boldsymbol{\phi}|\boldsymbol{z})q(\boldsymbol{z}|\boldsymbol{x}_n)\mathrm{d}\boldsymbol{z}$ and $p_0(\boldsymbol{\phi}) = \int p_0(\boldsymbol{\phi}|\boldsymbol{z})p_0(\boldsymbol{z})\mathrm{d}\boldsymbol{z} = \int p_0(\boldsymbol{\phi})p_0(\boldsymbol{z})\mathrm{d}\boldsymbol{z}$ (we assume priors are independent), according to our graphic model. The KL-term in formula (4): $\mathcal{D}_{KL}[p(\boldsymbol{\phi}|\boldsymbol{x}_n)||p(\boldsymbol{\phi})]$ can be factored into:

$$\mathcal{D}_{KL}[q(\boldsymbol{\phi}|\boldsymbol{x}_n)||p_0(\boldsymbol{\phi})] \tag{5}$$
$$= \int \Big[ q(\boldsymbol{\phi}|\boldsymbol{z})q(\boldsymbol{z}|\boldsymbol{x}_n)[\log q(\boldsymbol{\phi}|\boldsymbol{z}) + \log q(\boldsymbol{z}|\boldsymbol{x}_n) - \log p_0(\boldsymbol{\phi}) - \log p_0(\boldsymbol{z})] \Big] \mathrm{d}\boldsymbol{z} \mathrm{d}\boldsymbol{\phi}$$
$$= \int q(\boldsymbol{z}|\boldsymbol{x}_n)q(\boldsymbol{\phi}|\boldsymbol{z}) \frac{\log q(\boldsymbol{\phi}|\boldsymbol{z})}{\log p_0(\boldsymbol{\phi})} \mathrm{d}\boldsymbol{z} \mathrm{d}\boldsymbol{\phi} + \int q(\boldsymbol{\phi}|\boldsymbol{z})q(\boldsymbol{z}|\boldsymbol{x}_n) \frac{\log q(\boldsymbol{z}|\boldsymbol{x}_n)}{\log p_0(\boldsymbol{z})} \mathrm{d}\boldsymbol{z} \mathrm{d}\boldsymbol{\phi}$$
$$= \int q(\boldsymbol{z}|\boldsymbol{x}_n)\mathcal{D}_{KL}\Big[ q(\boldsymbol{\phi}|\boldsymbol{z})||p_0(\boldsymbol{\phi}) \Big] \mathrm{d}\boldsymbol{z} + \mathcal{D}_{KL}\Big[ q(\boldsymbol{z}|\boldsymbol{x}_n)||p_0(\boldsymbol{z}) \Big] \int q(\boldsymbol{\phi}|\boldsymbol{z})\mathrm{d}\boldsymbol{z} \mathrm{d}\boldsymbol{\phi} \tag{6}$$

The first KL-term in equation (6) constrains the mimic tree $\boldsymbol{\phi}$ to its prior when given the latent representation $p(Z|X)$. Intuitively, the unconditional prior of a mimic tree $\boldsymbol{\phi}$ is a tree with only root node. It is where our IB object encourages the simplify of mimic trees.

The second KL-term push the latent representation to its prior. In practice, the real posterior $p(\boldsymbol{z}|\boldsymbol{x})$ is intractable, and the variational inference framework allows us to approximate it with $q(\boldsymbol{z}|\boldsymbol{x}_n)$. Based on the VAE design, we add a reconstruction term that projects $Z$ back to $X$, in order to construct a complete VAE object. Since the expected log-likelihood $\mathbb{E}_{q(\boldsymbol{z}|\boldsymbol{x}_n)}[\log p(\boldsymbol{x}_n|\boldsymbol{z})] < 0$, the lower bound still holds, in fact, by maximizing this lower bound, we implicitly push this reconstruction term to zero, which improves the reconstruction performance.

By substituting Equation (6) back to Formula (4) and expanding it with a reconstruction term, we have:

$$I(\Phi; Y) - \lambda I(\Phi; X) \tag{7}$$

$$\geq \frac{1}{N} \sum_{n=1}^{N} \int \left[ q(\boldsymbol{\phi}|\boldsymbol{x}_n) \log q(y_n|\boldsymbol{\phi}) - \lambda \mathcal{D}_{KL}[q(\boldsymbol{\phi}|\boldsymbol{x}_n)\|p_0(\boldsymbol{\phi})] \right] \mathrm{d}\boldsymbol{\phi}$$

$$\geq \frac{1}{N} \sum_{n=1}^{N} \int \left\{ \mathbb{E}_{q(\boldsymbol{z}|\boldsymbol{x}_n)}[\log p(\boldsymbol{x}_n|\boldsymbol{z})] - \lambda \mathcal{D}_{KL}[q(\boldsymbol{z}|\boldsymbol{x}_n)\|p_0(\boldsymbol{z})] + \tag{8}$$

$$\mathbb{E}_{q(\boldsymbol{z}|\boldsymbol{x}_n)}\left[ q(\boldsymbol{\phi}|\boldsymbol{z}) \log q(y_n|\boldsymbol{\phi}) - \lambda \mathcal{D}_{KL}[q(\boldsymbol{\phi}|\boldsymbol{z})\|p_0(\boldsymbol{\phi})] \right] \right\} \mathrm{d}\boldsymbol{\phi}$$

$$= \frac{1}{N} \sum_{n=1}^{N} \left\{ \mathbb{E}_{q(\boldsymbol{z}|\boldsymbol{x}_n)}[\log p(\boldsymbol{x}|\boldsymbol{z})] - \lambda \mathcal{D}_{KL}[q(\boldsymbol{z}|\boldsymbol{x}_n)\|p_0(\boldsymbol{z})] + \tag{9}$$

$$\mathbb{E}_{q(\boldsymbol{z}|\boldsymbol{x}_n)}\left[ \int q(\boldsymbol{\phi}|\boldsymbol{z}) \log q(y_n|\boldsymbol{\phi}) \mathrm{d}\boldsymbol{\phi} - \lambda \mathcal{D}_{KL}[q(\boldsymbol{\phi}|\boldsymbol{z})\|p_0(\boldsymbol{\phi})] \right] \right\}$$

$$= \text{ELBo objective} + \text{ Expected VIB objective} \tag{10}$$

The proof is almost complete until here. Before going to the next step, We provide some intuitions about why we should use this lower bound as the training object. By maximizing this lower bound ( or the *ELBo object* and the *Expected VIB objective*), we essentially minimizing three measures: 1) $D_{KL}[p(y|\boldsymbol{\phi})\|q(y|\boldsymbol{\phi})]$ from formula (1), 2) $D_{KL}[p(\boldsymbol{\phi})\|p_0(\boldsymbol{\phi})]$ from formula (2), and 3) the negative expected log-likelihood $-\mathbb{E}_{q(\boldsymbol{z}|\boldsymbol{x}_n)}[\log p(\boldsymbol{x}_n|\boldsymbol{z})]$ together. The first and second measures come from Deep Variational Information Bottleneck [11]. The third measure is a reconstruction term that we added. This lower bound object shares many intuitions with the Evidence Lower Bound (ELBo) object in variational inference methods [12].

To derive a piratical object for tree learning, based on the intuitions from [13], the VIB object can be factored into a conditional two-stage Minimum Description Length (MDL) representation:

$$\frac{1}{N} \sum_{n=1}^{N} \mathbb{E}_{q(\boldsymbol{\phi}|\boldsymbol{z}_n)}[\log q(y_n|\boldsymbol{\phi})] - \lambda \mathcal{D}_{KL}[q(\boldsymbol{\phi}|\boldsymbol{z}_n)]\|p_0(\boldsymbol{\phi})]$$

$$= \frac{1}{N} \sum_{n=1}^{N} \mathbb{E}_{q(\boldsymbol{\phi}|\boldsymbol{z}_n)}\left[ \log q(y_n|\boldsymbol{\phi}) + \lambda \log p_0(\boldsymbol{\phi}) \right] + \lambda H[q(\boldsymbol{\phi}|\boldsymbol{z}_n)]$$

$$= \frac{1}{N} \sum_{n=1}^{N} -\mathbb{E}_{q(\boldsymbol{\phi}|\boldsymbol{z}_n)}\left[ \mathcal{L}_q(y_n) + \lambda \mathcal{L}_p(\boldsymbol{\phi}) \right] + \lambda H[q(\boldsymbol{\phi}|\boldsymbol{z}_n)]$$

$$= -\text{Conditional MDL} + \text{Entropy Regularizer} \tag{11}$$

where $\mathcal{L}_q(y_n)$ and $\mathcal{L}_p(\boldsymbol{\phi})$ denote the description length of encoding *target labels* $y_n$ with a mimic model $\boldsymbol{\phi}$ and encoding *the mimic model* $\boldsymbol{\phi}$ respectively. It gives the proof.

### B.3 Proof of Proposition 1

**proof.** Our proof is based on the conclusion of [13]: the cost of describing the structure of the tree with $N$ nodes and $L$ splits is $\mathcal{L}(N, L, (N+1)/2)$ and.

$$\mathcal{L}(N, L, \frac{(N+1)}{2}) = log((N+3)/2) + log\left(\binom{N}{L}\right) \text{bits} \tag{12}$$

The function $\mathcal{L}(N, L, \frac{(N+1)}{2})$ can be approximated using Stirling's approximation to obtain:

$$\mathcal{L}(N, L, \frac{N+1}{2}) = NH(L/N) + \frac{4\log(N)}{2} - \frac{\log(L)}{2} -$$
$$\frac{\log(N-L)}{2} - \frac{\log(2\pi)}{2} - \log(\frac{N+1}{2}) + \mathrm{O}(1/N)$$

In a binary regression tree, $N = 2L - 1$, so the cost of encoding tree structure is $\mathcal{L}(2L - 1, L, L)$, which gives the proof.

# C More Experiment Results

## C.1 Regression Performance

Table C.1 reports the complete regression performance as a complement to the experiment results in our main paper. We report the $mean \pm variance$ results for all the examined environments (including Flappy Bird, Space Invaders and Assault). The evaluation metrics include Root Mean Square Error (RMSE), Mean Absolute Error (MAE), Variance Reduction (VR), Variance Reduction Per-Leaf (VR-PL), and leaf numbers under. $w_+$ indicates that each leaf node contains a linear model whose parameter size equals the input data dimensions. We omit the linear model when computing VR, so VR evaluates only the splitting influence.

Table C.1: The *complete* regression performance.

| Game | Method | RMSE | MAE | VR | VR-PL | Leaf |
|---|---|---|---|---|---|---|
| Flappy Bird | CART | 1.69E-1 ± 4.87E-4 | 7.96E-2 ± 2.11E-5 | 8.51E-2 ± 1.46E-5 | 8.43E-5 ± 1.44E-11 | 1007 |
| | VIPER | 1.68E-1 ± 9.98E-4 | 7.66E-2 ± 5.13E-5 | 8.57E-2 ± 2.01E-5 | 1.88E-4 ± 9.03E-12 | 453 |
| | M5-RT | 1.05E-1 ± 1.01E-3 | 5.75E-2 ± 2.66E-5 | 9.59E-2 ± 1.73E-5 | 8.37E-5 ± 1.32E-11 | 1144 |
| | M5-MT | 8.49E-2 ± 1.11E-3 | 4.35E-2 ± 2.43E-5 | 9.56E-2 ± 1.69E-5 | 1.55E-4 ± 4.52E-11 | $612^{w+}$ |
| | GM-LMT | 1.63E-1 ± 1.25E-3 | 9.08E-2 ± 4.21E-5 | 8.99E-2 ± 1.57E-5 | 2.99E-4 ± 1.02E-11 | $303^{w+}$ |
| | VR-LMT | 2.56E-1 ± 1.21E-3 | 1.26E-1 ± 4.73E-5 | 8.46E-2 ± 1.31E-5 | 5.36E-4 ± 2.21E-11 | $157^{w+}$ |
| | VAE+CART | 1.98E-1 ± 5.79E-4 | 1.14E-1 ± 5.21E-5 | 7.25E-2 ± 1.46E-5 | 3.43E-4 ± 3.25E-10 | 212 |
| | VAE+VIPER | 1.80E-1 ± 5.53E-4 | 1.15E-1 ± 5.61E-5 | 7.63E-2 ± 3.31E-5 | 5.32E-4 ± 1.13E-10 | 143 |
| | VAE+GM-LMT | 1.88E-1 ± 6.32E-4 | 1.01E-1 ± 3.05E-5 | 6.35E-2 ± 5.91E-5 | 3.51E-4 ± 1.82E-9 | $180^{w+}$ |
| | VAE+VR-LMT | 1.42E-1 ± 5.97E-4 | 6.29E-2 ± 2.30E-5 | 7.95E-2 ± 1.77E-5 | 5.12E-4 ± 2.05E-9 | $154^{w+}$ |
| | VAE+MCRTS | 1.59E-1 ± 1.03E-3 | 7.98E-2 ± 7.53E-5 | 7.83E-2 ± 4.31E-5 | 1.27E-3 ± 7.23E-9 | 61 |
| | IDOE+CART | 1.37E-1 ± 2.53E-3 | 7.48E-2 ± 2.01E-5 | 8.23E-2 ± 3.81E-5 | 4.02E-4 ± 9.53E-10 | 204 |
| | IDOE+VIPER | 1.59E-1 ± 1.21E-3 | 6.44E-2 ± 5.67E-5 | 8.50E-2 ± 2.13E-5 | 4.48E-4 ± 6.73E-10 | 191 |
| | IDOE+GM-LMT | 1.38E-1 ± 5.17E-4 | 7.23E-2 ± 5.94E-5 | 7.87E-2 ± 3.16E-5 | 3.74E-4 ± 2.54E-10 | $212^{w+}$ |
| | IDOE+VR-LMT | 1.42E-1 ± 1.21E-3 | 7.02E-2 ± 6.12E-5 | 8.21E-2 ± 1.23E-5 | 7.16E-4 ± 8.94E-10 | $115^{w+}$ |
| | IDOE+MCRTS | 1.34E-1 ± 8.16E-4 | 7.54E-2 ± 5.12E-5 | 8.53E-2 ± 5.49E-5 | 1.37E-3 ± 6.40E-9 | 62 |
| Space Invaders | CART | 2.10E-1 ± 1.25E-4 | 1.09E-1 ± 2.22E-4 | 4.96E-2 ± 6.11E-7 | 7.02E-5 ± 1.23E-12 | 705 |
| | VIPER | 1.87E-1 ± 3.56E-4 | 1.12E-1 ± 1.28E-4 | 4.63E-2 ± 7.13E-6 | 8.80E-5 ± 6.18E-11 | 525 |
| | M5-RT | 2.18E-1 ± 9.37E-5 | 1.19E-1 ± 1.68E-4 | 4.54E-2 ± 2.52E-6 | 2.92E-5 ± 1.04E-12 | 1558 |
| | M5-MT | 1.88E-1 ± 4.70E-4 | 1.03E-1 ± 5.51E-4 | 1.60E-2 ± 2.21E-6 | 1.25E-5 ± 1.30E-12 | $1303^{w+}$ |
| | GM-LMT | 1.91E-1 ± 2.13E-4 | 1.18E-1 ± 2.01E-4 | 2.07E-2 ± 3.21E-6 | 8.32E-5 ± 1.31E-12 | $249^{w+}$ |
| | VR-LMT | 1.81E-1 ± 1.41E-4 | 1.17E-1 ± 2.77E-4 | 2.65E-2 ± 1.23E-6 | 1.61E-4 ± 2.14E-11 | $166^{w+}$ |
| | VAE+CART | 1.90E-1 ± 2.02E-4 | 1.08E-1 ± 3.62E-4 | 3.99E-2 ± 3.52E-6 | 7.86E-5 ± 1.37E-11 | 507 |
| | VAE+VIPER | 1.89E-1 ± 3.13E-4 | 1.12E-1 ± 2.01E-4 | 4.12E-2 ± 5.12E-5 | 9.89E-5 ± 7.13E-10 | 417 |
| | VAE+GM-LMT | 1.96E-1 ± 2.31E-4 | 1.15E-1 ± 2.56E-4 | 3.39E-2 ± 1.32E-5 | 2.75E-4 ± 7.68E-10 | $123^{w+}$ |
| | VAE+VR-LMT | 1.98E-1 ± 1.17E-4 | 1.13E-1 ± 2.01E-4 | 3.52E-2 ± 4.13E-5 | 2.08E-4 ± 1.43E-9 | $171^{w+}$ |
| | VAE+MCRTS | 1.89E-1 ± 3.54E-4 | 1.16E-1 ± 3.95E-4 | 4.82E-2 ± 1.34E-5 | 5.66E-4 ± 1.02E-9 | 85 |
| | IDOE+CART | 1.91E-1 ± 4.31E-4 | 1.05E-1 ± 2.04E-4 | 5.21E-2 ± 3.12E-4 | 1.38E-4 ± 7.16E-10 | 375 |
| | IDOE+VIPER | 1.79E-1 ± 5.01E-4 | 1.11E-1 ± 1.13E-4 | 5.26E-2 ± 4.12E-4 | 1.69E-4 ± 1.23E-10 | 313 |
| | IDOE+GM-LMT | 1.82E-1 ± 1.64E-4 | 1.17E-1 ± 5.17E-4 | 4.79E-2 ± 5.17E-5 | 3.23E-4 ± 5.23E-9 | $149^{w+}$ |
| | IDOE+VR-LMT | 1.73E-1 ± 4.24E-4 | 1.02E-1 ± 3.21E-4 | 4.54E-2 ± 7.26E-7 | 3.79E-4 ± 9.14E-11 | $120^{w+}$ |
| | IDOE+MCRTS | 1.64E-1 ± 7.63E-5 | 1.14E-1 ± 3.52E-4 | 5.37E-2 ± 6.14E-6 | 7.08E-4 ± 2.13E-9 | 76 |
| Assault | CART | 2.59E-1 ± 3.93E-3 | 9.30E-2 ± 7.29E-5 | 4.79E-2 ± 5.05E-4 | 7.46E-5 ± 1.22E-9 | 642 |
| | VIPER | 2.22E-1 ± 4.77E-5 | 1.16E-1 ± 4.64E-5 | 5.28E-2 ± 3.97E-5 | 8.09E-5 ± 2.12E-10 | 653 |
| | M5-RT | 3.06E-1 ± 3.97E-3 | 1.27E-1 ± 1.06E-4 | 4.37E-2 ± 1.47E-4 | 2.73E-5 ± 5.71E-11 | 1605 |
| | M5-MT | 2.28E-1 ± 3.07E-3 | 1.05E-1 ± 3.14E-5 | 3.42E-2 ± 2.93E-5 | 2.54E-5 ± 1.60E-11 | $1351^{w+}$ |
| | GM-LMT | 1.27E-1 ± 3.29E-3 | 8.72E-2 ± 4.89E-5 | 5.55E-2 ± 2.27E-4 | 1.83E-4 ± 2.40E-9 | $307^{w+}$ |
| | VR-LMT | 1.95E-1 ± 4.56E-3 | 9.35E-2 ± 1.32E-4 | 5.80E-2 ± 2.11E-4 | 1.98E-4 ± 2.49E-9 | $291^{w+}$ |
| | VAE+CART | 2.51E-1 ± 4.10E-3 | 1.10E-1 ± 6.23E-5 | 5.15E-2 ± 6.39E-4 | 1.16E-4 ± 3.19E-9 | 448 |
| | VAE+VIPER | 2.45E-1 ± 3.03E-3 | 1.20E-1 ± 5.75E-5 | 4.57E-2 ± 6.76E-4 | 1.29E-4 ± 7.12E-9 | 356 |
| | VAE+GM-LMT | 2.01E-1 ± 3.89E-3 | 1.31E-1 ± 7.05E-5 | 4.20E-2 ± 6.38E-4 | 1.44E-5 ± 4.11E-9 | $293^{w+}$ |
| | VAE+VR-LMT | 1.75E-1 ± 4.10E-3 | 1.19E-1 ± 2.43E-4 | 5.10E-2 ± 3.08E-4 | 1.99E-4 ± 4.63E-9 | $258^{w+}$ |
| | VAE+MCRTS | 1.86E-1 ± 3.66E-3 | 1.23E-1 ± 7.00E-5 | 6.58E-2 ± 1.85E-4 | 7.75E-4 ± 2.36E-7 | 85 |
| | IDOE+CART | 2.03E-1 ± 3.23E-3 | 1.05E-1 ± 6.21E-5 | 5.67E-2 ± 2.56E-4 | 1.81E-4 ± 6.01E-9 | 315 |
| | IDOE+VIPER | 1.81E-1 ± 4.12E-3 | 1.06E-1 ± 7.15E-5 | 6.05E-2 ± 2.54E-4 | 1.90E-4 ± 7.01E-9 | 319 |
| | IDE+GM-LMT | 1.79E-1 ± 3.19E-3 | 1.21E-1 ± 7.63E-5 | 5.45E-2 ± 6.18E-4 | 2.15E-4 ± 8.91E-9 | $256^{w+}$ |
| | IDE+VR-LMT | 1.69E-1 ± 3.16E-3 | 1.15E-1 ± 8.57E-5 | 6.03E-2 ± 3.14E-4 | 2.27E-4 ± 5.61E-9 | $268^{w+}$ |
| | IDE+MCRTS | 1.65E-1 ± 5.01E-3 | 1.15E-1 ± 7.25E-5 | 7.53E-2 ± 4.27E-4 | 9.07E-4 ± 4.12E-7 | 83 |

## C.2 Leaf-By-Leaf Regression Performance

We report the plot for the leaf-by-leaf regression performance with the RMSE and MAE metrics. The leaf number runs from 1 to 30. For CART, VIPER, GM-LMT, and VR-LMT, we apply post-pruning to constrain their leaf numbers.

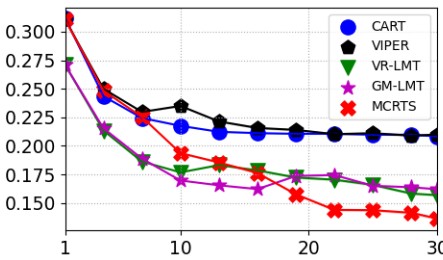

Figure C.1: Leaf-by-leaf RMSE in the Flappybird environment.

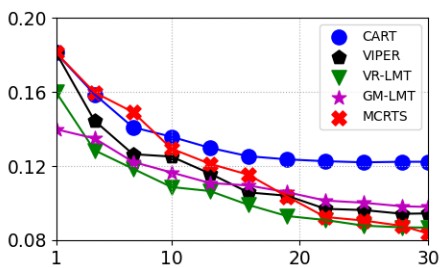

Figure C.2: Leaf-by-leaf MAE in the Flappybird environment.

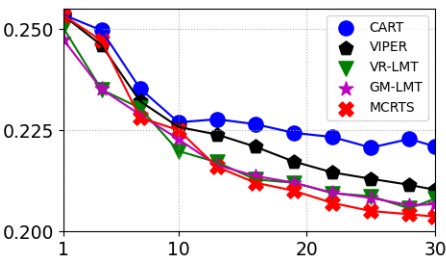

Figure C.3: Leaf-by-leaf RMSE in the Space Invaders environment.

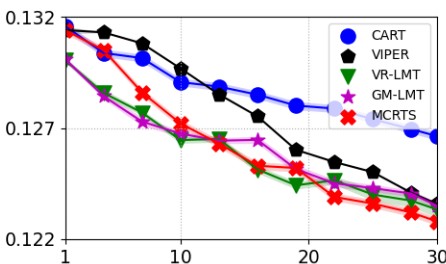

Figure C.4: Leaf-by-leaf MAE in the Space Invaders environment.

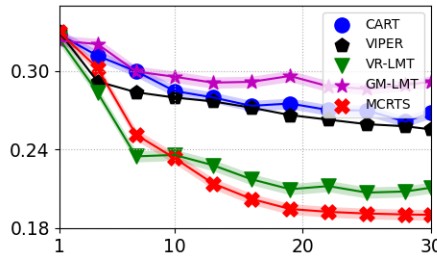

Figure C.5: Leaf-by-leaf RMSE in the Assault environment.

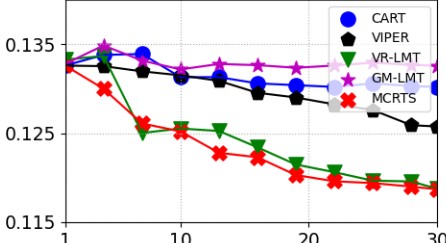

Figure C.6: Leaf-by-leaf MAE in the Assault environment.

## D    Human Evaluation

In the human evaluation experiment, we interviewed a total of 12 participant. The survey was conducted blindly, so the participants were not given any context information regarding the baseline models or our approach. Each participant was provided a short presentation explaining : 1) the background of interpretable DRL, including the motivation and the goal. 2) A brief introduction of interpreting DRL methods in the flappy bird environment, including saliency map, super pixels, and object representations from IDOE. After the presentation, they were asked two multiple-choice questions, one that is about recognizing the physical meaning of the extracted feature in each method. (Check the question in Figure D.1, Figure D.2, Figure D.3, and Figure D.4) and the other one is about ranking these methods (Check questions in Figure D.5). Table D.2 and Table D.1 show a summary of responses we received.

Table D.1: Number of the votes for each feature including: Feature (1): distance between the bird and the pillar, Feature (2): the location of the bird, Feature (3): the location of the pillar, Feature (4): relative position between the bird and pillars, and Feature (5): the shape (width or length) of the pillar.

| Method | Feature (1) | Feature (2) | Feature (3) | Feature (4) | Feature (5) |
|---|---|---|---|---|---|
| Saliency Map | 5 | **7** | 4 | **7** | 2 |
| Super Pixels | 3 | **4** | 5 | 0 | 0 |
| Object Representation | **12** | 7 | 6 | **12** | **12** |

Table D.2: Number of the votes for the rankings.

| Method | Best | Mideum | Worst |
|---|---|---|---|
| Saliency Map | 2 | 10 | 0 |
| Super Pixels | 0 | 0 | 12 |
| Mimic Tree | 10 | 2 | 0 |

**Saliency Map** are perturbation-based method:
1) Randomly perturb the input space
2) Measure the difference of output before and after perturbation.
2) Mask the important region (with blue color).

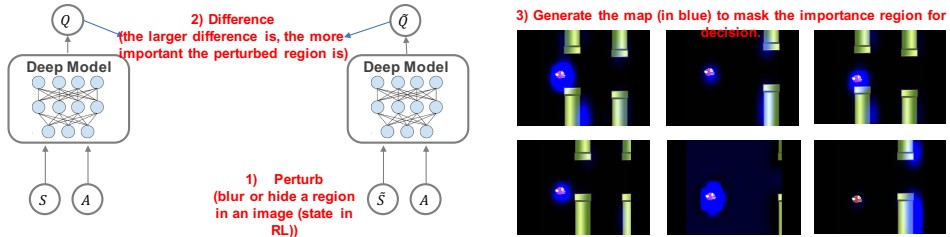

**Question 1**: Can you select which has been captured by these maps:
(1) distance between the bird and the pillar. (2) the location of the bird (3) the location of the pillar (4)relative position between the bird and pillars  (5) the shape (width or length) of the pillar.

Figure D.1: Human evaluation page 1.

**Mimic Learning**: Train a decision tree with the same input (action and state (image of size 3*128*128) and soft-outputs (logistics) from neural network.
**Super-pixels** are the pixels in the splitting node of a tree

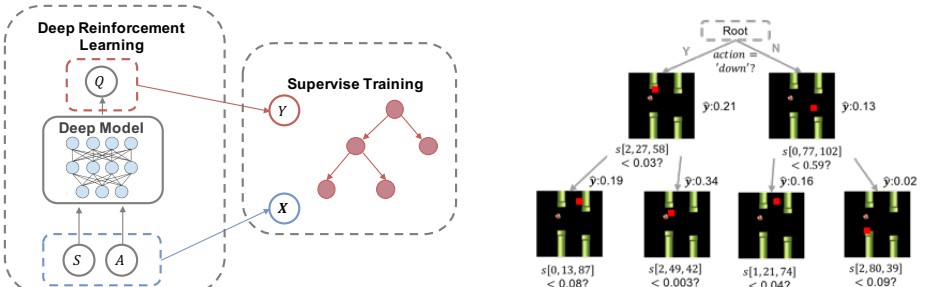

**Question 2**: Can you select which has been captured by these pixels:
(1) distance between the bird and the pillar. (2) the location of the bird (3) the location of the pillar (4)relative position between the bird and pillars  (5) the shape (width or length) of the pillar.

Figure D.2: Human evaluation page 2.

**Latent representation** learns a symbolic abstraction of state (input image):
1) detect objects from RL states,
2) capture the variation of each object with latent variables with VAE.
3) interpret the variables with latent traversal.

**Step 1**: Detect Object (check **white** parts)

Input Image     Object 0     Object 1     Object 2

Background     pillars     Bird and part of a pillar

**Step 2**: Capture Object variation with latent Variables.

**Step 3**: Interpret the variables with latent traversal.

For example, to interpret $Z_{2,1}$ (the 1st variable for the 2nd object)
1) Fix the value of other variables (e.g., $Z_{i \neq 2, j \neq 1} = 0$).
2) Traverse $Z_{2,1}$ by setting it from its min to max value.
3) For each value, we generate an image by the VAE decoder.
4) Observe the generate images (e.g., in the first line on the right plot) and conclude what has been captured.

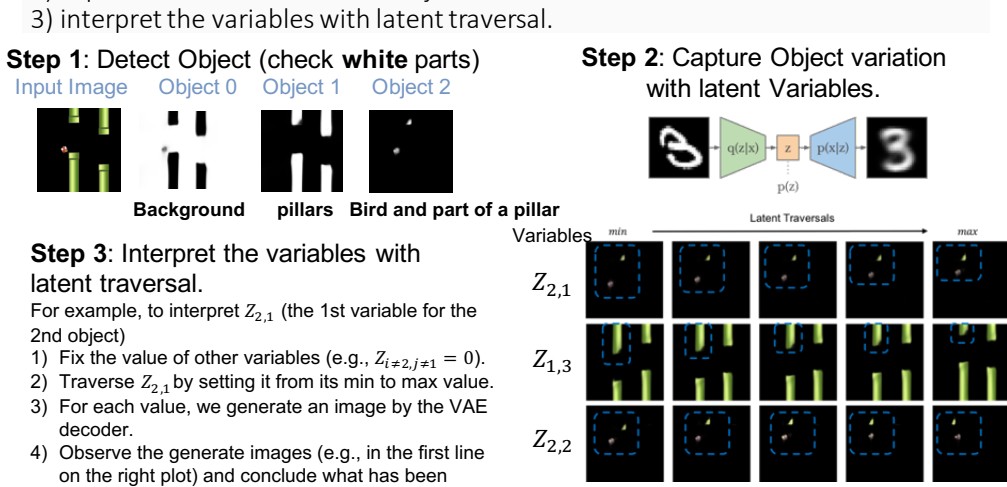

Figure D.3: Human evaluation page 3.

**Latent representation** learns a symbolic abstraction of state (input image):
1) detect objects from RL states,
2) capture the variation of each object with latent variables with VAE.
3) interpret the variables with latent traversal

**Step 3**: Interpret the variables with latent traversal.

For example, to interpret $Z_{2,1}$ (the 1st variable for the 2nd object)
1) Fix the value of other variables (e.g., $Z_{i \neq 2, j \neq 1} = 0$).
2) Traverse $Z_{2,1}$ by setting it from its min to max value.
3) For each value, we generate an image by the VAE decoder.
4) Observe the generate images (e.g., in the first line on the right) and conclude what has been captured.

$Z_{2,1}$
$Z_{1,3}$
$Z_{2,2}$

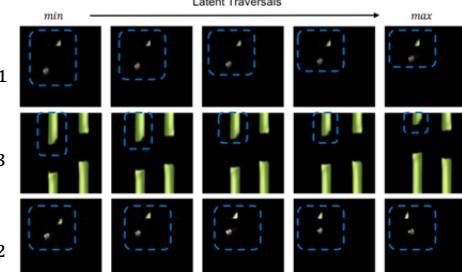

**Question 3**: Can you select which has been captured by these traversals? (1) distance between the bird and the pillar. (2) the location of the bird (3) the location of the pillar (4)relative position between the bird and pillars (5) the shape (width or length) of the pillar.

Figure D.4: Human evaluation page 4.

**Background**: Different RL interpretations:

**2. Super-pixels (global explanation):**
1) Build a tree with **all samples**
2) Based on raw inputs, large tree size (over 1000 nodes)
3) Highlight pixels in the decision path.
4) Conclude the knowledge from the super-pixels.

**3. Mimic Tree (global explanation):**
1) Build a tree with **all samples**
2) Based on latent features, smaller tree size (less than 100 nodes)
3) Visualize decision rules and the influence of a split by the decoder.

Question 4: Which Interpretation you prefer? Please rank them

Figure D.5: Human evaluation page 5.

# E    More Examples

## E.1    Examples of VAE Factorization

We independently train two VAEs. Each VAE learns a latent representation for the states in the flappy bird environment. To better visualize the results, we set the latent dimension to 5 ($D = 5$). Figure E.1 shows the latent traversals for latent variables learned by these VAEs. It shows the VAEs capture inconsistent variations, and thus it is hard to use the latent variables from one VAE to identify the latent variables learned by another VAE.

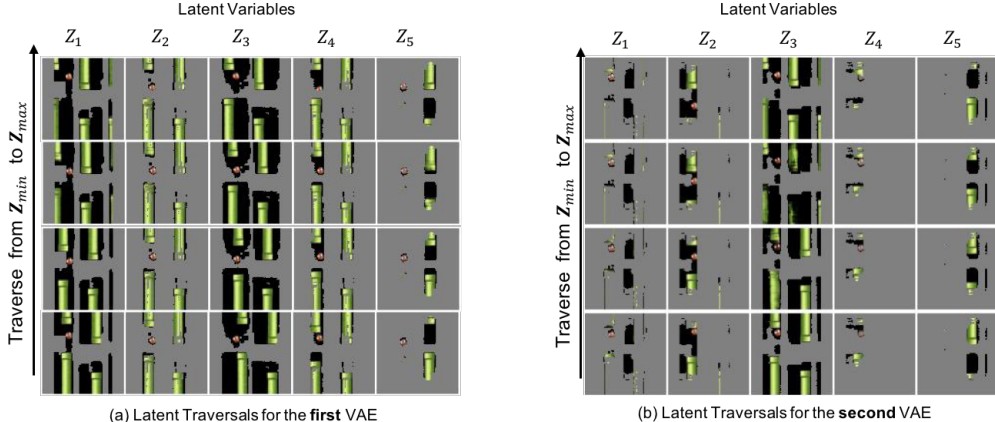

(a) Latent Traversals for the **first** VAE          (b) Latent Traversals for the **second** VAE

Figure E.1: The latent traversals for the latent variables learnt by two VAEs.

## E.2    An Examples of MCRTS

To further clarify the description about MCRTS (section 5.2 in our main paper), we illustrate an example of extracting mimic trees from a path from the MCRTS search tree in Figure E.2.

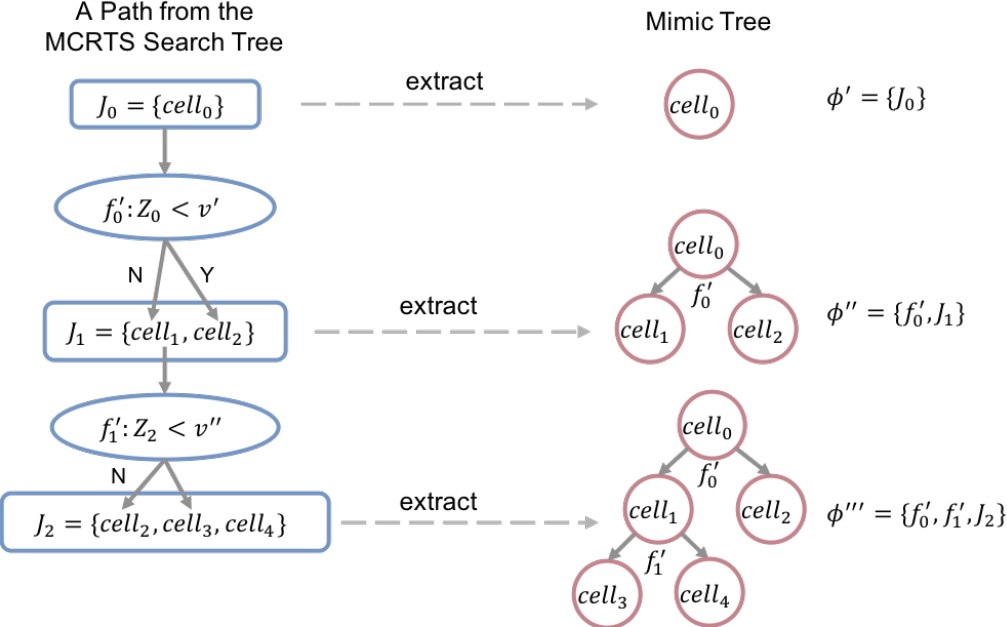

Figure E.2: An example of extracting mimic trees from a path from the MCRTS search tree. Note our MCRTS constructs a *search tree* where an edge represent a split in the extracted *mimic tree*.

### E.3 Examples of I-MONets

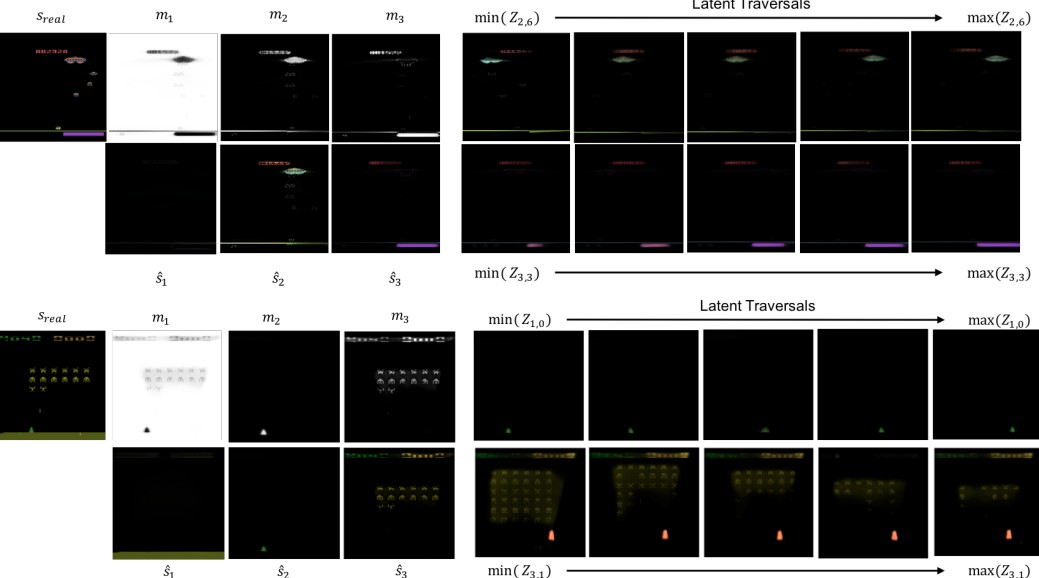

Figure E.3: Visualizations for IMONet outputs in the Assault (Upper) and Space Invaders (Left) environments. IMONet decomposes a state $s_{real}$ into three objects with masks $m_1$ (for the background), $m_2$ and $m_3$, where *white/dark* colors mark *captured/uncaptured* regions. The generations from the decoder are $\hat{s}_1$, $\hat{s}_2$, and $\hat{s}_3$.

## F Ethics Statement

We expect the major ethical impact of our work to be in Explainable AI (XAI). XAI is one of the most important approaches to building a trustworthy AI system with transparent and predictable behavior. Such efforts are valuable and have supports from our governments. For example, the European Parliament approved a General Data Protection Regulation (GDPR) which gives individuals the right to request "meaningful information of the logic involved" when automated decision-making takes place with "legal or similarly relevant effects" on individuals. From the perspective of XAI, the positive and negative societal implications of our work are as follows:

**Positive Outcomes.** We build a robust XAI system for DRL models. The system provides more intuitive and concise explanations, compared to previously-purposed DRL interpretations. Such explanations make the black-box DRL models (for sequential decision making) understandable. Their predictable behavior will substantially increase the trust of human-users. A variety of DRL downstream applications, including the self-driving car and recommendation system, can potentially benefit from our inventions.

**Negative Outcomes.** The advance of XAI encourages the government to endorse laws that regulate the level of transparency in AI systems. This effort, however, increases the difficulty of developing new AI systems and might slow down the boosting AI industry in recent years.