# OpenReview forum: "Learning Tree Interpretation from Object Representation for Deep Reinforcement Learning"
_NeurIPS.cc/2021/Conference — NeurIPS 2021 Poster_

### Official Review · Reviewer_zwwU · 2021-07-08

**Rating:** 4
**Confidence:** 3

**Summary:**

The paper proposed a new explanation approach for reinforcement learning.

**Ethical Concerns:**

No ethical concern

**Limitations And Societal Impact:**

The authors have adequately addressed the limitations and potential negative societal impact of their work.

**Main Review:**

The research is a little bit incremental. There are already many interpretation methods in the context of reinforcement learning. The work proposed is yet another method that generates explanation in higher fidelity. However, the evaluation of higher fidelity interpretation is insufficient. Some existing techniques are not compared (see below).

In addition to the methods based on interpretable trees, there are many other explanation methods for RL applications (e.g., saliency map-based approaches and attention-based approaches). The paper does not conduct an experimental comparison with these alternative approaches. It would be helpful to show the performance difference between alternative approaches and the proposed method.

To evaluate the effectiveness of the proposed method, the authors measure the fidelity of the model's capability in approximation. This, to some extent, indicates fidelity, but it does not imply a good mimicking performance is a good explanation. I am happy to see the authors conduct counterfactual analysis to assess the fidelity of interpretability. However, the authors only show a case but do not quantify the fidelity. I think it would be more convincing if the authors could follow existing and commonly adopted methods to assess the fidelity of the interpretation (e.g., blinding some critical component in the explanation and see how that influences the action of the target agent quantitatively).

The paper shows some interesting results on some simple games. I wonder if the proposed method is applied to more sophisticated games (e.g., starcraft or super Mario that have a more complicated background and foreground), will the proposed method still generate a decent latent representation for background and objects? It would be more promising if the authors could show results on more sophisticated games.

I am happy to see the authors evaluate the explainability by conducting a user study. First, I am not sure if the authors received IRB approval before conducting this human study. Second, the human study is subjective. Even if an interpretation makes sense to a human, it does not imply it is the right explanation. Can the authors justify the validity of this human-based assessment?

**Time Spent Reviewing:**

2.5 hours

---

> ### Author Response · Authors · 2021-08-10
> **Response to Reviewer zwwU**
>
> Dear Reviewer,
>
> We appreciate your constructive comments and provide our response as follows.
>
> 1. " *The research is a little bit incremental. There are already many interpretation methods in the context of reinforcement learning.*"
>
> * **Response** We believe there is plenty of space for developing a more advanced interpretation method for RL. Existing methods are commonly based on low-level input features, whereas this work explores learning tree interpretations based on object-level features. It is a less-explored direction in which we incorporate advanced representation learning techniques into RL interpretation.
>
> 2. " *In addition to the methods based on interpretable trees, there are many other explanation methods for RL applications (e.g., saliency map-based approaches and attention-based approaches). The paper does not conduct an experimental comparison with these alternative approaches.* "
>
> * **Response** We have compared saliency map-based approaches and attention-based approaches from multiple perspectives, including complexity, consistency, and interpretability. For more details, please check 1) illustrative examples (lines 336-345), 2) human evaluation (lines 346-360) and 3) discussion of model complexity (Section 6.3).
>
> 3. " *I am happy to see the authors conduct counterfactual analysis to assess the fidelity of interpretability. However, the authors only show a case but do not quantify the fidelity.* "
>
> * **Response** Section 6.1 has studied the fidelity of our interpretation. The complete results are in Table C.1 (in appendix), which provides a quantification of the degree of fidelity.  The RMSE and MAE columns quantify the degree of discrepancy between the decisions computed by each mimic tree and the neural network that it approximates.
>
> 4. " *The paper shows some interesting results on some simple games. I wonder if the proposed method is applied to more sophisticated games (e.g., starcraft or super Mario that have a more complicated background and foreground), will the proposed method still generate a decent latent representation for background and objects?* "
>
> * **Response** We agree that starcraft is one of the most challenging RL games. Developing a controlling and interpretation method in this environment will be hard. However, for Super Mario, it is a well-explored game in Object-Oriented RL (see lines 187-193). A previous work [1] has shown that the agent can solve this game by modeling the interactions of objects. We believe IDOE can help to learn a mimic policy for this game as well.
>
> [1] Mohan, Shiwali, and John Laird. "An object-oriented approach to reinforcement learning in an action game." Proceedings of the AAAI Conference on Artificial Intelligence and Interactive Digital Entertainment. Vol. 6. No. 1. 2011.
>
> 5. " *human study is subjective. Even if an interpretation makes sense to a human, it does not imply it is the right explanation. Can the authors justify the validity of this human-based assessment?* "
>
> * **Response** The human evaluation is conducted to evaluate interpretability instead of correctness (see motivation in line 346 and lines 368-369). To study the correctness of the explanation, we evaluate the fidelity of our mimic interpretation in Section 6.1. We also demonstrate the correctness of some interpretations by showing that they are consistent with those from the saliency map-based approaches and attention-based approaches (lines 342 - 344). Apart from this, we empirically explain how the interpretations match some common sense (line 345). They together suggest the correctness of our interpretation, after which we conduct a human evaluation to study interpretability.

---

### Official Review · Reviewer_nsW5 · 2021-07-15

**Rating:** 6
**Confidence:** 4

**Summary:**

This paper proposes a novel method of interpreting individual decisions of a DRL policy trained by deep Q learning. The explanation they want to deliver is to identify the important features within an input state that contribute most to the corresponding Q function output. Technically, it first learns an encoding layer to transform the high-dimensional input into low-dimensional hidden representations. To enable explainability, the encoding layer is designed to preserve the physical meaning of the objects in raw inputs. Using the hidden space representations, it proposes a Monte Carlo Regression Tree Search (MCRTS) algorithm to train a tree to mimic the action advantages.  To validate the effectiveness of the proposed model, the authors evaluate the proposed technique from two aspects: fidelity and interpretability. The evaluation on three RL environments demonstrates the superiority of the proposed method over some of the existing explanation methods.


**Limitations And Societal Impact:**

The authors discussed two limitations in Section 6.3. Additional limitations and future works could be (1) The proposed technique can only be applied to environments using images as inputs; (2) Generalizability of the proposed technique to more complicated environments, multi-agent environments; (3) Generalizability of the proposed technique to policy networks.

I appreciate that the authors include a social impact discussion. And I agree with the discussion points. To make it more comprehensive, the authors could discuss the possible negative impact from the adversarial point of view, .e.g, attackers may leverage the proposed explanation methods to develop stronger attacks.

**Main Review:**

This paper is overall well organized and written. The proposed technique is interesting and clear explained. I like the idea of the two-step interpretation: hidden representation + mimic model, and I appreciate the authors' effort in keeping the identifiability and interpretability of the latent representations. They are very important properties towards providing explanations. The evaluations are relatively well designed and comprehensive. I appreciate that the authors include a relatively complete comparison with the baselines. IMHO, the paper would be more convincing if the authors could address my following comments.

My main concern is in the fidelity evaluation. I appreciate that the authors evaluate this important property. However, the current evaluation is not comprehensive. More specifically, approximation accuracy is essential for mimic-based explanations. IMHO, it is only the first step towards a comprehensive fidelity evaluation. More importantly, I would suggest the authors evaluating whether the derived explanations actually reflect the decision-making process of the neural networks. The authors could refer to some existing fidelity evaluation metrics used in the papers that try to derive important features as explanations[1,2]. The basic idea of these metrics is to perturbing the identified features and observing the changes in the corresponding NN's outputs. If the identified features are truly important ones, perturbing them will trigger a large change to the output. The authors could borrow this idea and design a fidelity metric to evaluate the explanation fidelity or just use the existing ones (If they are applicable). In summary, enabling an accurate approximation is only the first step towards achieving faithful explanations. The authors need to conduct another evaluation directly attached to the derived explanations.

Some minor comments:

1. I may miss this detail. I am wondering whether the MCRTS can be applied to the raw features. If yes, I would like to see the results. It could further demonstrate the effectiveness of the feature extraction step.

2. The current version only showcases the explanations on one of the three environments. I would suggest demonstrating the explanations on the other two applications and conducting the human evaluation on those applications too.

3. The proposed technique utilizes some existing methods to address some of the challenges. Among them, VAE is a well-known technique, which may not need an introduction in this work. However, [34] may not be that well-known. I would appreciate that if the authors could provide some background knowledge regarding these non-well-known techniques.

======== Post rebuttal Comments===============

After reading the authors' response, my concern about fidelity evaluation still stands. I would still suggest designing a quantitative fidelity evaluation w.r.t. the RL agents together with an evaluation metric. The authors could design their own metric by following the methodology similar to the counterfactual analysis (remove and rerun). With the metric, the authors could compare the fidelity of their proposed approach with that of existing explanation methods. As things stand, I would be inclined to maintain my current score.


**Time Spent Reviewing:**

6 hours

---

> ### Author Response · Authors · 2021-08-09
> **Response to Reviewer nsW5**
>
> Dear Reviewer,
>
> We appreciate your constructive comments and provide our response as follows.
>
> 1. " *I would suggest the authors evaluating whether the derived explanations actually reflect the decision-making process of the neural networks. The authors could refer to some existing fidelity evaluation metrics used in the papers that try to derive important features as explanations[1,2].*"
>
> * **Response:** Section 6.2 studies the interpretability of explanations where we have compared our explanation to previous perturbation-based methods. When it comes to the correctness of this explanation, 1) we include a counterfactual analysis in Figure 5 (There is a typo in Figure 5c, 0.23 in the bottom node should be 0.15. ).  This method is similar to perturbation-based methods. By fixing a latent feature with the $do$ operator,  we perturb the object captured by this feature. The influence of this perturbation can be observed by comparing the mimic tree output before and after the perturbation (0.23 v.s., 0.15). We will expand this experiment and include more examples in the revised version. 2) We also demonstrate the correctness of interpretation by showing it is consistent with that from the saliency map-based approaches and attention-based approaches (lines 342 - 344).
>
> 2. " *I am wondering whether the MCRTS can be applied to the raw features. If yes, I would like to see the results. It could further demonstrate the effectiveness of the feature extraction step.*"
>
> * **Response:** Launching MCRTS on raw pixels is very inefficient and in most cases computationally intractable, so we did not apply MCRTS on raw features. We discussed briefly this issue in Section 4.1 (lines 140 - 142). MCRTS runs Monte Carlo simulations that scan through candidate splitting values at each tree level, which is computationally more expensive than greedy tree-building algorithms (see limitation in Section 6.2).
>
> 3. " *The current version only showcases the explanations on one of the three environments. I would suggest demonstrating the explanations on the other two applications and conducting the human evaluation on those applications too.*"
>
> * **Response:** We have learned mimic trees in the other two environments, but the visualization and case studies are removed due to space limitations. We will place these results in the appendix in the revised version.
>
> 4. " *Among them, VAE is a well-known technique, which may not need an introduction in this work. However, [34] may not be that well-known. I would appreciate that if the authors could provide some background knowledge regarding these non-well-known techniques.*"
>
> * **Response:** We agree that more words should be spent on introducing [34]. In the revised version, we will add more details about [34] to make our paper more self-contained.
>
> 5. " *Limitations And Societal Impact*"
>
> * **Response:** Thanks for your suggestion, we will include the limitations and potential societal impacts in the revised version.

---

> > ### Comment · Reviewer_nsW5 · 2021-08-16
> > **Responses to the authors.**
> >
> > Thanks the authors for responding to my comments. Regarding the fidelity evaluation, the authors responded by pointing to Sec. 6.2, where a set of evaluations about explanations were conducted. I appreciated the authors' efforts on those evaluations. However, IMHO, those experiments still cannot fully evaluate the explain fidelity w.r.t the original RL systems and agents. This is mainly because of two reasons (1) The counterfactual experiment reflects the fidelity w.r.t. to the mimic tree but not the original RL agents. It is possible that the explanations could correctly reflect the mimic trees but not the original RL agents even though the approximation is accurate. (2) Regarding the second argument given by the authors, recent works have shown that attentions can give misleading explanations [1,2]. More generally, different explanation methods can give different explanations with different fidelities.  As such, using the explanations generated by existing methods as references for fidelity evalution may not be that rigorous.  In summary, given the current experiments, I still suggest designing and conducting a quantitative fidelity evaluation w.r.t. the ***original RL agents***. The authors could design their own metric by following the Methodology similar to the counterfactual analysis, but taking the ***original RL agents*** as the object. With the metric, the authors could compare the fidelity of their proposed approach with that of existing explanation methods.
> >
> > [1] Attention is not Explanation.
> > [2] Attention is not not Explanation.

---

> > > ### Author Response · Authors · 2021-08-16
> > > **Thanks for the comments and response to your concerns**
> > >
> > > Thanks for your suggestions.
> > >
> > > * We agree that conducting a counterfactual experiment that directly studies the influences of perturbation on the **original RL agents** is necessary. We have integrated this step into our counterfactual analysis. To be more specific: a) after fixing the latent variable (e.g., set $Z_{3,1}=0$), we apply the IDOE decoder that decodes the perturbed latent features back to images (lines 331-332). These images are similar to the **perturbed images** in the regular perturbation-based methods. b) we ask the original RL agents to generate action values based on the perturbed images (lines 332-333). This is an **interaction with the original RL agents**. If we stop here (without learning the mimic tree), we can study the influence of this perturbation (the do operator) by comparing the action values computed with the original images (without applying the do operator) and perturbed images (apply the do operator). We will expand our counterfactual analysis by adding this intermediate step.
> > >
> > > * We agree that *"different explanation methods can give different explanations with different fidelities"*. In fact, due to the lack of identifiability, the same method can generate different explanations with the same fidelity.  This is why [1][2] (or [5][6] in our paper) showed inconsistency between attention and feature importance: very different attention distributions can yield equivalent predictions. Their works offered us the motivation of studying the model identifiability.
> > > In our experiment section, we include results from the attention-based methods as a comparison to our model and empirically show that one can find the consistency between different interpretations. We agree this observation should not be rigorous evidence of fidelity and will clarity this in the revised version.

---

### Official Review · Reviewer_8RF7 · 2021-07-19

**Rating:** 6
**Confidence:** 3

**Summary:**

This paper focuses on building a tree representation from pixels to evaluate interpretability in deep reinforcement learning. The proposed approach relies on features learned by a variational autoencoder. A mimic tree then extracts disentangled object representations for showing the causal impact. The method is evaluated on three game environments: Flappy Bird, Space Invaders and Assaults.

**Ethical Concerns:**

No general concern.

**Limitations And Societal Impact:**

Authors mention “A potential negative societal impact of our work is that it encourages
the government to endorse laws that regulate the level of transparency in AI systems. This may
increase the difficulty of deploying advanced AI systems.” as a negative impact which can be seen as positive in most cases.


**Main Review:**

Strength:
The paper is well-written. Increasing interpretability in learned features for RL is very useful for real-world tasks. The methodology and experiments are well explained.
Weakness:
* Line 18: as far as I know vision RL still remains a challenging task except some known environments such as atari. It would be useful if authors cite works outperforming human level control.
* Line 188: Authors mention that one can collect enough data to learn object representation in RL. This in real world scenarios is not applicable and is the main reason researchers focus on sample efficiency.
* Line 185: Can authors provide more details on how unet was used for attention network? This seems to make it computationally expensive.


**Time Spent Reviewing:**

2

---

> ### Author Response · Authors · 2021-08-09
> **Response to Reviewer 8RF7**
>
> Dear Reviewer,
>
> We appreciate your constructive comments and provide our response as follows.
>
> 1. " *Line 18: as far as I know vision RL still remains a challenging task except some known environments such as atari. It would be useful if authors cite works outperforming human level control."*
>
> * **Response:** We will revise this paper by explicitly citing the environment where RL agents have achieved human-level control (e.g., Atari Mojuco, Dota2, and Starcraft).
>
> 2. " *Line 188: " Authors mention that one can collect enough data to learn object representation in RL. This in real world scenarios is not applicable and is the main reason researchers focus on sample efficiency."*
>
> * **Response:** We agree that in practice sample efficiency is important for RL, and that's why many model-based RL methods have been proposed in recent years for reducing the interactions between agent and environment, but in this work, we focus on interpreting RL, especially the value functions in the model-free methods (not planning). Since the agent is in an online learning environment that allows exploration, our algorithm exploits this setting a little for receiving sufficient samples from environments. How to design an efficient exploration algorithm will be a promising future direction.
>
> 3. " *Line 185: Can authors provide more details on how unet was used for attention network? This seems to make it computationally expensive."*
>
> * **Response:** Attention network is trained to generate spatial masks for object detection. The input/output of U-Net are images and spatial masks. U-Net is implemented by the layer-by-layer convolutional neural network and instance normalization, followed by a sigmoid function that transfers logistics to attention. Compared to VAE, adding attention will increase model complexity, but this increase of computational complexity is not significant since we train CVAE and attention network together with an EBLo loss. We will add more details to our revised version.

---

> > ### Comment · Reviewer_8RF7 · 2021-08-14
> > **Read author response**
> >
> > Thanks for your response.
> > I keep the same rating.

---

### Official Review · Reviewer_SaJC · 2021-07-20

**Rating:** 6
**Confidence:** 3

**Summary:**

The paper proposes learning decision trees over latent features using MCRTS to improve interpretability of Deep Reinforcement Learning.

The paper leverages the Information Bottleneck principle to find explainable trees with minimum description length.

The proposed method uses VAE to learn a disentangled representation on which to build regression trees that more easily interpreted.

**Ethical Concerns:**

It is not clear if proper ethical guidelines were followed while doing the human evaluation, for example, how the participants were chosen and how well informed they were.

**Limitations And Societal Impact:**

The paper briefly mention some limitations of the work in section 6.3, although focus on computational complexity rather than interpretability complexity. It is not clear if the proposed method increase a shallow interpretability of RL.

The claim that "A potential negative societal impact of our work is that it encourages the government to endorse laws that regulate the level of transparency in AI systems." has no base, and also it is not clear it would be negative societal impact, instead of positive.

**Main Review:**

# Originality:
The problem of understanding and being able to interpret DNN used for RL is an important problem, both for developing and validating new ideas, as for increasing the safety and confidence on the learnt models.

The paper introduces a novel combination of VAEs with MCTRS to learn a more interpretable policy that can mimic the behavior of the DNN.

# Quality:
It is not clear how IDOE is capable of learning an identifiable disentangling object encoder, and how identifiable and disentangled are the variables and whether they correspond to objects or not.

Missing comparisons with Gradient Boosting Trees.

There is no validation of the claim that VAEs learn disentangled representations, or that the latent variables are interpretable by humans.

There is not proof that the latent variables learnt by VAEs are more interpretable than DNNs latent features.

In Section 4.2 it's unclear that the latent representation can capture objects and its relations in the different tasks.

In section 4.4 the claim that IDOE learns a symbolic abstraction of state space by representing object variations is not sustained. Showing 2 cherry picked examples are enough.

It is not clear how the number of objects and latent size was chosen and those impact the results.

The human evaluation with 12 participants is very weak. The paper didn't present a good protocol for human evaluation. For instance not clear how the participant were chosen and interviewed.

# Clarity:
The is not very clear about its main contributions and its significance. The writing is not very clear or well organized.

For example section 4.3 is confusing on how the model is implemented and how the different CVAE are defined or learnt.

Not explanation of why Flappy Bird, space Invaders and Assaults where chosen for the experiments.

The Figure 5 representing the decision tree is hard to interpret the changes on the outputs. For example how to read the  advantage of different actions (down vs up) on different branches.

# Significance:
The paper claims a promising mimic performance, but not clear how good is the mimic policy at actually solving the tasks.

The paper shows some interesting and promising results, but its applicability to complex RL tasks is missing.


# Minor issues:
In Figure 4, only the right figure has confidence bands around the different algorithms.

**Time Spent Reviewing:**

5 hours

---

> ### Author Response · Authors · 2021-08-09
> **Response to Reviewer SaJC**
>
> Dear Reviewer,
>
> We appreciate your constructive comments and provide our response as follows.
>
> 1. "*It is not clear how IDOE is capable of learning an identifiable disentangling object encoder, and how identifiable and disentangled are the variables and whether they correspond to objects or not.*"
>
> * **Response**: The identifiability of IDOE is evaluated by the MCC metric (Section 4.2, lines 163-171). MCC is a well-studied metric for evaluating the identifiability of representations (see reference [33] in our paper). For the disentangling ability, IDOE applies a factored prior, and thus the KL-divergence in Equation (3) can explicitly encourage disentangled representations. These properties have been carefully studied in a VAE (see [39] in our paper). Our IDOE is implemented by a VAE-based encoder and trained with the ELBo loss, so its disentangling ability is obvious. We also use illustrative examples (in Figure 2) to demonstrate the disentangling performance of VAE.
>
> 2. "*There is no validation of the claim that VAEs learn disentangled representations, or that the latent variables are interpretable by humans. There is not proof that the latent variables learnt by VAEs are more interpretable than DNNs latent features.*"
>
> * **Response**: We have explained how IDOE derives disentangled representations in our previous response. We demonstrate and evaluate interpretability from multiple perspectives (Section 6.2), including 1) the illustrative examples to show the interpretability of object representation (Figure 2) and the mimic tree from MCRTS (Figure 5) 2) the human evaluations to empirically quantify the level of interpretability. Potential limitations are discussed in Section 6.3.
>
> 3. "*In section 4.4 the claim that IDOE learns a symbolic abstraction of state space by representing object variations is not sustained. Showing 2 cherry picked examples are [not] enough.*"
>
> * **Response**: We have studied three environments. Flappy bird is selected as an illustrative example for introducing our methodology.  We will include additional examples for the other two environments in supplementary material in the revised version.  Note that the human evaluation was designed precisely to address this issue by providing a more comprehensive and reliable evaluation of our approach.
>
> 4. "*It is not clear how the number of objects and latent size was chosen and those impact the results.*"
>
> * **Response**: We list the experiment details in the appendix (Section A). The hyper-parameters are mostly determined empirically with the validation datasets. We will include the experiment details about the tested candidate hyper-parameters and the corresponding performance in the examined RL environments.
>
> 5. "*The human evaluation with 12 participants is very weak. The paper didn't present a good protocol for human evaluation. For instance not clear how the participant were chosen and interviewed.*"
>
> * **Response**: The details of human evaluation, including the format, questions, and results, are moved to Section D (in our appendix). The participants are volunteers from four Canadian universities. The participants are interviewed by a short presentation that explains: 1) the background of interpretable DRL, including the motivation and the goal. 2) A brief introduction to interpreting DRL methods, including saliency map, superpixels, and object representations from IDOE.
>
> 6. "*The is not very clear about its main contributions and its significance.*"
>
> * **Response**: The main contributions of our paper can be summarized as:
>      1. We propose a RAMi framework that enables representing the raw input with our novel IDOE and searching for the optimal mimic tree with the MCRTS algorithm.
>      2. We derive an information-theoretic IB-MDL objective that incorporates both the fidelity and simplicity for mimic tree learning.
>      3. We introduce our method of leveraging the mimic tree to extract causal relations from a DRL model.
> We will provide a clear summarization of these contributions in our revised version.
>
> 7. "*Not explanation of why Flappy Bird, space Invaders and Assaults where chosen for the experiments.*"
>
> * **Response**: This paper is aimed at interpreting RL algorithms, and a majority of the existing RL algorithms are evaluated on the benchmark of gaming environments (e.g, Atari). This is the main reason why we study these environments.
>
> 8. "*The paper briefly mention some limitations of the work in section 6.3, although focus on computational complexity rather than interpretability complexity. It is not clear if the proposed method increase a shallow interpretability of RL.*"
>
> * **Response**:  We are not sure what the reviewer means by "shallow interpretability". If this refers to the complexity of interpretability, our paper mentioned that our model jointly optimizes both simplicity and fidelity of the mimic tree, which significantly reduces the complexity of interpretation.
>
> 9. "*Missing comparisons with Gradient Boosting Trees.*"
>
> * **Response**:  We are not aware of any work that uses GBT for interpretability, since 1) GBT is an unparametric ensemble model that builds multiple trees (forest) to improve prediction instead of interpretability. The number of parameters grows dramatically during training, and GBT is likely to be more complex than a neural network.  2) Most GBTs are mainly for classification rather than regression. Developing an RL interpretation by integrating the knowledge from multiple decision trees (or forests) will be hard.

---

> > ### Comment · Reviewer_SaJC · 2021-08-25
> > **RE: Response to Reviewer SaJC**
> >
> > I want to thank the authors for their detailed response and clarifications, I have updated my score accordingly.

---

### Decision · Program_Chairs · 2021-09-28

**Decision:**

Accept (Poster)

**Comment:**

This paper explores the problem of learning an interpretable version of the decision procedure used by neural RL, which reviewers agree is interesting. There was some disagreement about how substantial the results are, especially regarding fidelity and practicality. While the majority of reviewers ended up mildly positive, I believe this paper can be substantially improved by taking the comments to heart in guiding further experiments.

**Consistency Experiment:**

NeurIPS has a long history of experimentation. In 2014, NeurIPS ran an experiment in which 10% of submissions were reviewed by two independent committees to quantify the randomness in the review process. This year, we repeated a variant of this experiment to see how the quality of the review process has changed over time.  This paper was part of the experiment and was therefore assigned to two committees (consisting of reviewers, an Area Chair, and a Senior Area Chair) that reached independent decisions.  If both committees made the same recommendation, this recommendation was followed. If a single committee recommended acceptance, the paper was accepted (with the exception of a few cases in which the other committee identified what we considered a fatal flaw, e.g., an error in a key result).

This copy’s committee reached the following decision: **Reject**

The other committee assigned to the paper recommended **Accept (Poster)**.  You can find the other set of reviews, along with any follow up discussion with the authors here:
https://openreview.net/forum?id=jb5fp_wQGHU